# NEURAL DIVERSITY REGULARIZES HALLUCINATIONS IN SMALL LANGUAGE MODELS

## ABSTRACT

Language models continue to hallucinate despite increases in parameters, compute, and data. We propose *neural diversity* — decorrelated parallel representations — as a principled mechanism that reduces hallucination rates at fixed parameter and data budgets. While existing mitigation strategies largely target accuracy, we provide the first formal tail bounds for hallucination probability in ensembled language models, reframing it as a second-moment reliability problem and *explaining 96.2% of empirical reliability variation* seen across parallel configurations. We introduce ND-LoRA (Neural Diversity Low-Rank Adaptation), combining parallel LoRA adapters with Barlow Twins regularization, and *reduce hallucinations by up to 25.6% (and 14.6% on average)* while preserving general accuracy. Ablations show LoRA adapters and regularization act synergistically, causal interventions prove neurodiversity as the mediating factor and correlational studies indicate scale: a 0.1% neural correlation increase is associated with a 3.8% hallucination increase. Finally, task-dependent optimality emerges: different tasks require different optimal amounts of neurodiversity. Together, our results highlight neural diversity as a third axis of scaling — orthogonal to parameters and data — to *improve the reliability of language models at fixed budgets*.

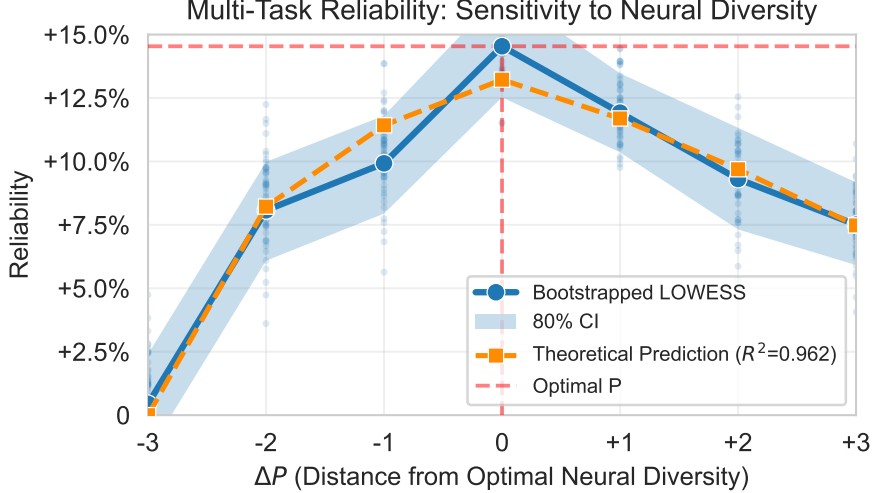

Figure 1: **Maximizing reliability requires optimal neural diversity.** Varying the number of decorrelated parallel representations $P \in \{1, 2, 4, 8\}$ across 6 hallucination benchmarks (182,850 samples, LOWESS, 80% CI), we find a U-shaped curve where performance peaks at optimal $P_\star$ ($\Delta P = P - P_\star$) then degrades. Theorems 1 & 2 predict this precisely ($R^2 = 0.962$, orange line) — explaining 96.2% of empirical reliability variation — and enable principled architectural design (ND-LoRA) that reduces hallucinations by 14.6% on average without degrading general capabilities.

| Category | Task | Best $P_\star$ | Best Score | $\Delta\%$ Score | Sig. |
|---|---|---|---|---|---|
| Hallucination | HaluEval (Dialog) | 4 | 0.516 | +12.8% | *** |
| | HaluEval (QA) | 4 | 0.451 | +23.4% | *** |
| | HaluEval (Summ) | 4 | 0.502 | +25.6% | *** |
| | MemoTrap v2 | 8 | 0.689 | +8.8% | *** |
| | TruthfulQA (MC1) | 2 | 0.269 | +7.3% | |
| | TruthfulQA (MC2) | 2 | 0.442 | +9.5% | * |
| Knowledge | NQ (8-shot) | 1 | 0.066 | – | |
| | NQ-swap | 8 | 0.554 | +0.8% | |
| | PopQA | 1 | 0.111 | – | |
| | TriviaQA (8-shot) | 1 | 0.192 | – | |

Table 1: **Optimal neural diversity is task-dependent: hallucination tasks benefit from neural diversity, knowledge tasks do not.** De-aggregating Figure 1, hallucination benchmarks consistently show large gains with increased diversity (up to 25.6%, HaluEval-Summ, $P_\star = 4$), while knowledge retrieval mostly peaks at $P_\star = 1$. This asymmetry supports hallucination as a reliability problem distinct from factual recall. Significance: *** $p < 0.001$, * $p < 0.05$.

# 1 INTRODUCTION

Despite scaling to trillions of parameters, language models hallucinate persistently (Lin et al., 2021). This reliability crisis is acute for small language models — increasingly favored for edge and agentic use cases (Zheng et al., 2025; Belcak et al., 2025) — whose compressed representations make them especially vulnerable to hallucinations, with even well-resourced efforts like GPT-OSS 20B exhibiting 91% hallucination rates on factual benchmarks (OpenAI, 2025).

Current hallucination mitigation strategies are largely empirically driven but theoretically ungrounded and target average performance rather than tail risk. RLHF optimizes mean harmlessness (Bai et al., 2022), RAG improves average factual grounding (Niu et al., 2024), and contrastive decoding enhances mean generation quality (Li et al., 2023b). While inference-time approaches like self-consistency and LoRA ensembling (Wang et al., 2022; 2023) reduce hallucinations through diverse sampling, they lack formal tail-probability guarantees. Similarly, parallel scaling methods (Chen et al., 2025) target first-moment improvements in perplexity and task accuracy. Yet controlling catastrophic failures requires bounding the tails of $\mathbb{P}(\text{hallucination})$, not just optimizing mean behavior.

Formal ensemble theory exists but targets the wrong objective. Classical ensemble methods (Krogh & Vedelsby, 1994) provide rigorous diversity theory to reduce mean generalization error $\mathbb{E}[\text{loss}]$, not tail-probability bounds for hallucinations. Deep ensembles (Lakshminarayanan et al., 2017) quantify uncertainty but lack hallucination-specific guarantees. Without explicit diversification, parallel architectures suffer *representational collapse* (Jing et al., 2022), leaving reliability gains unrealized.

To our knowledge, we provide the first formal framework for **hallucination probability tail bounds in ensembled language models**, reframing it as a second-moment reliability problem. Drawing on portfolio theory (Markowitz, 1952), we prove that decorrelated parallel representations (*neural diversity*) reduce this tail bound and introduce **ND-LoRA (Neural Diversity Low-Rank Adaptation)** to concretely demonstrate its hallucination reduction capabilities.

Our contributions are:

- **Theoretical Linkage**: We reframe hallucinations as a second-moment reliability problem and prove (i) a portfolio-theoretic bound showing hallucination probability $\mathbb{P}(\text{H}) \propto 1/P$ with $P$ decorrelated parallel representations (Theorem 1); and, (ii) non-monotonicity in reliability scaling (Theorem 2), showing that excessive parallelism can degrade diversity (and thus reliability) under common circumstances. We further show (iii) our theoretical predictions achieve $R^2 = 0.962$ in fitting empirical reliability gains (Figure 1), establishing quantitative validation rare in neural hallucination research.

- **Constructive Demonstration**: We demonstrate empirical gains via ND-LoRA (parallel LoRA + Barlow Twins decorrelation), which reduces hallucinations by up to 25.6% (and

14.6% on average) at $1.00004\times$ continued pretraining cost while preserving general capabilities across 12 benchmarks (Table 1, 2).

- **Mechanistic Analysis**: We establish that neural diversity mediates hallucination in four ways: (i) causality via perturbation ($p < 0.001$, Table 3), (ii) quantitative scale via correlation ($+0.1\%$ diversity $\Leftrightarrow$ $-3.8\%$ hallucination, Figure 3), (iii) super-linear effects via ablation (Table 4), and (iv) task-dependent optima via scaling sweeps (Table 1).

Neural diversity represents a third scaling axis beyond parameters and data. While traditional scaling asks "how big?" and data scaling "how much?", diversity scaling asks "how different?" — crucial for achieving reliability without massive computational investment.

## 2 A THEORY OF NEURAL DIVERSITY

Why don't existing scaling methods improve reliability? Without explicit diversity mechanisms, gradient descent drives parallel streams toward similar representations through *representational collapse* (Jing et al., 2022), leaving reliability gains unrealized. We establish the first hallucination tail bounds for ensembled language models, proving that neural diversity reduces hallucinations and providing mathematical foundations for ND-LoRA.

Our strategy adapts portfolio theory to neural architecture design. Classical ensemble methods reduce *mean error* $\mathbb{E}[\text{loss}]$ through variance reduction (Krogh & Vedelsby, 1994), treating correlation as a factor that limits accuracy gains. In contrast, portfolio theory manages *tail risk* — rare but catastrophic failures — by diversifying across correlated assets (Markowitz, 1952). We adapt the latter framework to tail bound hallucination probability $\mathbb{P}(\text{hallucination})$, where correlation becomes the primary control variable for reliability rather than a secondary constraint on mean performance.

### 2.1 PRELIMINARIES

Modern language models hallucinate by fabricating facts, generating content inconsistent with input, or creating unsupported claims (Maynez et al., 2020; Ji et al., 2023). While comprehensive taxonomies exist (Huang et al., 2024), we model hallucinations through a simple signal-noise proxy that captures the underlying reliability failure (Figure 1) while remaining analytically tractable.

**Signal-noise model.** Let $x \in X$ be a query with oracle output $y_\star(x) \in \mathbb{R}^V$ and corresponding hidden representation $z_\star(x) \in \mathbb{R}^d$. Consider an architecture that employs $P$ parallel computational pathways called *streams*, each processing the same input $X$ through the same model but in perturbed ways. We model the hidden output of each stream as $Z_i = z_\star + \varepsilon_i$ where $\varepsilon_i \in \mathbb{R}^d$ is centered noise with variance $\sigma_i^2 > 0$. The noise covariance $\Sigma \in \mathbb{R}^{P \times P}$ has entries $\Sigma_{ij} \triangleq \mathbb{E}[\langle \varepsilon_i, \varepsilon_j \rangle]$ with pairwise correlations $\rho_{ij} \triangleq \Sigma_{ij}/(\sigma_i \sigma_j)$ for $i \neq j$. We aggregate hidden representations via $\widehat{Z}_w = \sum_i w_i Z_i$ with weights summing to one. For readability, we omit $x$ where obvious and denote the average noise variance by $\bar{\sigma}^2 \triangleq \mathbb{E}[\sigma_i^2]$ and average correlation $\bar{\rho} \triangleq \mathbb{E}_{i<j}[\rho_{ij}]$.

**High-dimensional structure.** High-dimensional representations exhibit predictable geometric regularity that we exploit for analysis. We assume: (i) *Lipschitz decoding*, where outputs $\widehat{Y}_w(x) = f(\widehat{Z}_w(x))$ and $y_\star(x) = f(z_\star(x))$ satisfy $\|f(z) - f(z')\|_2 \leq L\|z - z'\|_2$ for some $L > 0$; and, (ii) *norm concentration*, where $\|\tilde{z}_i(X)\|_2^2 \approx d$ with small relative variance for per-feature whitened representations $\tilde{z}_i$. Both properties are standard in high-dimensional probability (Vershynin, 2018) and neural network analysis (Fazlyab et al., 2019; Bartlett et al., 2017).

**Neural representations.** At a chosen design layer, each stream exposes a $d$-dimensional representation $z_i(X)$. We whiten per-feature to obtain $\tilde{z}_i$ with zero mean and identity covariance. For streams $i < j$, the cross-correlation matrix is $C^{(ij)} \triangleq \mathbb{E}\left[\tilde{z}_i \tilde{z}_j^\top\right] \in \mathbb{R}^{d \times d}$ whose diagonal entries measure same-feature similarity and off-diagonal entries capture cross-feature alignment. Finally, using the widely-exploited observation that trained networks exhibit locally linear behavior at their operating point (Goodfellow et al., 2015; Simonyan et al., 2014), we connect representations to noise via local linearity: $\xi_i = A\tilde{z}_i$ for a shared linear readout $A \in \mathbb{R}^{d \times d}$ with finite condition number $\kappa$.

**Neural diversity index.** We define a simple cosine-based index to measure cross-stream diversity:

$$\mathcal{D} \triangleq \sqrt{\mathop{\mathbb{E}}_{i<j}\left[\frac{(\tilde{z}_i \cdot \tilde{z}_j)^2}{\|\tilde{z}_i\|^2 \|\tilde{z}_j\|^2}\right]}. \tag{1}$$

Lower $\mathcal{D}$ indicates greater neural diversity: $\mathcal{D} = 0$ means all streams are perfectly orthogonal, while $\mathcal{D} = 1$ means streams have suffered complete collapse.

**Hallucinations.** We define the output error as $E_w \triangleq \|\widehat{Y}_w(x) - y_\star(x)\|_F$, which is comparable to metrics like TruthfulQA-MC2 (Lin et al., 2021). For tolerance $\delta > 0$, the *hallucination event* is $H_\delta \triangleq \{E_w \geq \delta\}$. Our goal is to bound $\mathbb{P}(H_\delta)$ as a function of neural diversity $\mathcal{D}$ across streams $P$.

## 2.2 Neural Diversity Bounds Hallucination

Classical portfolio theory (Markowitz, 1952) gives the variance of an equally weighted portfolio of $P$ assets with average variance $\bar{\sigma}^2$ and average pairwise correlation $\bar{\rho}$ as:

$$\mathrm{Var}(Y) = \bar{\sigma}^2 \left(\frac{1-\bar{\rho}}{P} + \bar{\rho}\right). \tag{2}$$

To use this observation for hallucinations, we must first connect neuron-level representations to portfolio-level noise correlations. Exploiting the fact that (i) our ensemble has one underlying model with aligned neuron-level representations and (ii) our model has geometric regularity in representation and output, the following lemma establishes this mapping:

**Lemma 1** (Average Correlation Bound). *Suppose there exists a constant $C_4 \geq 1$ such that $\mathbb{E}[\|\xi_i\|_2^4] \leq C_4\, \sigma_i^4$ for all $i$. Then the average pairwise noise correlation satisfies*

$$|\bar{\rho}| \leq C_* \mathcal{D}, \tag{3}$$

*where $C_* \triangleq \sqrt{C_4}\, \kappa^2$ depends only on the kurtosis bound and the readout condition number $\kappa$.*

*Proof sketch.* We proceed in two steps: *(1)* Spectral bounds imply linear readout distorts cosines by at most $\kappa^2$, so noise-space diversity $\mathcal{D}_\xi \leq \kappa^2 \mathcal{D}$. *(2)* Cauchy–Schwarz twice — inner products to cosines, then kurtosis — gives $|\rho_{ij}| \leq \sqrt{C_4}\, \mathcal{D}_{\xi,ij}$; averaging pairs completes the proof. $\square$

We now have a direct path to tail-bound $P(H_\delta)$ as a function of $\mathcal{D}$ and $P$. For readability, we assume uniform weights $w_i = 1/P$ below but our approach can also be easily applied to arbitrary weights.

**Theorem 1** (Hallucination Bound with Diversity). *For any tolerance $\delta > 0$,*

$$\mathbb{P}(H_\delta) \leq \frac{\frac{1-C_* \mathcal{D}}{P} + C_* \mathcal{D}}{\frac{1-C_* \mathcal{D}}{P} + C_* \mathcal{D} + SNR}, \tag{4}$$

*where $SNR \triangleq \delta^2/\bar{\sigma}^2$.*

*Proof sketch.* Lemma 1 bounds $|\bar{\rho}| \leq C_* \mathcal{D}$, linking noise variance to representational diversity. Plugging into Equation 2, applying Chebyshev and normalizing by $\bar{\sigma}^2$ yields the stated bound. $\square$

This completes the first half of our theoretical result: Neural diversity mediates hallucination probability. With perfect de-correlation ($\bar{\rho} = 0$), hallucination probability scales as $O(1/P)$ — more streams reduce hallucination risk. When streams collapse ($\bar{\rho} = 1$), the bound becomes independent of $P$, explaining why naive ensembling without diversification provides no reliability benefits.

## 2.3 Non-Monotonic Scaling Behavior

Next, we demonstrate that under common circumstances, the hallucination bound follows a U-shaped curve — initially decreasing with higher $P$, but starts increasing eventually. Consider the case where the correlation itself increases with $P$, say, due to optimizer constraints:

**Theorem 2** (U-shaped Behavior). *Suppose $\bar{\rho}(P) = \rho_0 + \beta(P-1)^\gamma$ for constants $\rho_0 \in [0,1)$, $\beta > 0$, $\gamma > 0$. Define*

$$v(P) \triangleq \mathrm{Var}(E_w) = \bar{\sigma}^2 \left( \frac{1-\bar{\rho}(P)}{P} + \bar{\rho}(P) \right), \qquad B(P) \triangleq \frac{v(P)}{v(P)+\delta^2}. \tag{5}$$

*Then $B(P)$ is U-shaped: there exists $P_\star \geq 1$ minimizing $\mathbb{P}(\mathrm{H}_\delta)$, with $P_\star$ controlled by how fast $\bar{\rho}(P)$ degrades with $P$.*

*Proof sketch.* The hallucination bound $B(P)$ is monotonic in variance $v(P)$, so we analyze $v(P)$ directly. There are two competing effects: the $1/P$ term drives variance down, while growing correlation $\bar{\rho}(P) = \rho_0 + \beta(P-1)^\gamma$ eventually dominates. Differentiating shows $v'(P)$ changes sign exactly once, yielding a unique minimum $P_\star$ whose location depends $\beta, \gamma$ and $\rho_0$. $\qquad \square$

This theorem establishes *non-monotonicity* — hallucination probability $\mathbb{P}(\mathrm{H}_\delta)$ actually *increases* for larger $P$, meaning reliability degrades. This is stronger than the well-known diminishing returns of ensembles (where improvement slows but continues). While ensemble theory also shows optimal size matches the number of class labels for accuracy-optimized classifiers (Bonab & Can, 2019), we prove and validate (Figure 1) that diversity can degrade in generative language models with excessive parallelism under common circumstances and also harm reliability.

## 2.4 THEORETICAL VALIDATION

By measuring empirical diversity $\mathcal{D}(P)$ and plugging these values into Theorem 1's bound, we achieve $R^2 = 0.962$ (Figure 1), explaining 96.2% of empirical reliability variation. This fit uses only two free parameters ($C_\star, SNR$) shared across all tasks and observations, with $\mathcal{D}(P)$ fixed from empirical measurements. Theorem 2's correlation growth model provides a mechanism for observed concavity: correlation grows as $O((P-1)^\gamma)$, overwhelming the $O(1/P)$ diversification benefit. This alignment — rare in hallucination research where theory often lags empirics — validates our portfolio-theoretic framework.

Together, Theorem 1 and Theorem 2 show that (i) reducing $\mathcal{D}$ reduces hallucinations and (ii) there exists an optimal $P_\star$ that minimizes hallucinations. Next, we show how to construct an architecture and training protocol to reduce $\mathcal{D}$ and find $P_\star$.

## 3 ND-LoRA: A PRACTICAL DEMONSTRATION

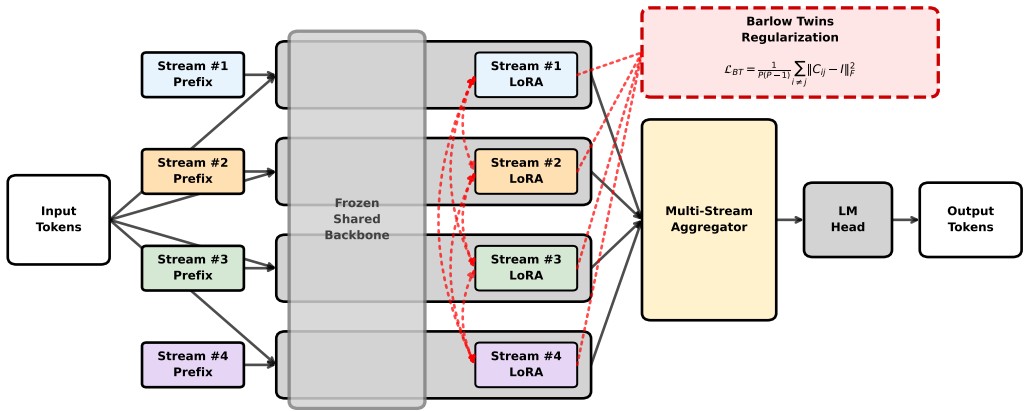

Figure 2: **ND-LoRA schematic for $P = 4$ parallel streams.** Each stream receives independent LoRA adapters and learnable prefix tokens. The aggregator combines stream outputs with learnable weights, while Barlow Twins regularization incentivizes decorrelation between stream outputs.

We introduce ND-LoRA (Neural Diversity Low-Rank Adaptation), a parameter-efficient method that demonstrates our theoretical framework for neural diversity regularization. ND-LoRA extends the ParScale architecture with stream-aware LoRA adapters and explicit decorrelation objectives. Figure 2 visually summarizes our approach.

## 3.1 ARCHITECTURE

Our implementation builds on ParScale with $P$ parallel computation streams. Each stream $i \in \{1, \ldots, P\}$ uses 48 learnable prefix tokens prepended to the input sequence that flow through all layers via the attention mechanism, along with stream-specific LoRA adapters applied at each layer:

$$h_i^{(\ell)} = \text{Layer}^{(\ell)}(h_i^{(\ell-1)} + B_i^{(\ell)} A_i^{(\ell)} h_i^{(\ell-1)}) \tag{6}$$

where $B_i^{(\ell)} \in \mathbb{R}^{d \times r}$, $A_i^{(\ell)} \in \mathbb{R}^{r \times d}$ are stream-specific LoRA matrices with rank $r$. The final output combines streams through a learned aggregator:

$$y = \text{LM\_Head}\left(\sum_{i=1}^{P} w_i \cdot h_i^{(L)}\right) \tag{7}$$

where $w_i = (1 - \varepsilon) \cdot \text{softmax}(\text{MLP}([h_1^{(L)}, \ldots, h_P^{(L)}]))_i + \varepsilon/P$ are dynamic weights with label smoothing ($\varepsilon = 0.1$) computed from the concatenated stream representations. This prevents attention collapse by ensuring minimum weight $\varepsilon/P$ for each stream.

This architecture enables stream specialization while maintaining parameter efficiency. For $P = 2$ streams with rank-16 LoRA, we use approximately 29K trainable parameters per layer, comparable to a single rank-32 LoRA but with fundamentally different representational capabilities.

## 3.2 BARLOW TWINS REGULARIZATION

To encourage neural diversity, we apply Barlow Twins regularization across all pairs of streams $i < j$ at a pre-specified design layer $\ell_\star$.

Let $z_i \in \mathbb{R}^{B \times T \times d}$ denote the hidden representations of stream $i$ at the design layer for a batch of size $B$ and sequence length $T$. We first apply batch normalization and mean-centering to obtain whitened features $\tilde{z}_i$. We then calculate the cross-correlation matrices $C^{(ij)} \in \mathbb{R}^{d \times d}$ as in subsection 2.2 and apply standard Barlow Twins (Zbontar et al., 2021) for each pair of streams $i < j$:

$$\mathcal{L}_{BT} = \mathop{\mathbb{E}}_{i<j} \left\| C^{(ij)} - I \right\|_F \tag{8}$$

The total training objective combines cross-entropy and decorrelation terms:

$$\mathcal{L} = \mathcal{L}_{CE} + \lambda_{BT} \mathcal{L}_{BT} \tag{9}$$

# 4 EXPERIMENTAL VALIDATION

We validate ND-LoRA through systematic hallucination reduction experiments using parameter- and data-matched comparisons. We describe our full experimental setup in subsection A.5.

## 4.1 KEY RESULTS

Table 2 demonstrates ND-LoRA achieves substantial improvements on hallucination-sensitive benchmarks while maintaining competitive general performance. ND-LoRA with $P = 2$ streams achieves statistically significant improvements on HaluEval-Summarization (0.481* vs 0.400, $p < 0.001$, 8.1% absolute / 20.2% relative), TruthfulQA-MC2 (0.442* vs 0.403, $p = 0.030$, 3.9% absolute / 9.5% relative) and MemoTrap (0.666* vs 0.634, $p < 0.001$, 3.2% absolute / 5.1% relative) vs parameter-matched Qwen, validating our theoretical prediction.

Although ND-LoRA's improvements specifically target reliability benchmarks, they preserve general capabilities. Qwen slightly outperforms on Wikitext (0.778 vs. 0.784) and Natural Questions (0.065 vs. 0.055), but ND-LoRA wins slightly on Winogrande (0.574 vs. 0.572).

| Model | HaluEval | MemoTrap | TruthfulQA | NQ | Wikitext | WG |
|---|---|---|---|---|---|---|
| ND-LoRA R16 (P=2) | **0.481*** | **0.666*** | **0.442*** | 0.055 | 0.784 | **0.574** |
| ParScale R32 (P=2) | 0.439 | 0.638 | 0.412 | 0.059 | 0.793 | 0.564 |
| Qwen LoRA R32 | 0.400 | 0.634 | 0.403 | **0.065** | **0.778** | 0.572 |

Table 2: **Even at $P = 2$ streams, ND-LoRA achieves up to 20.2% relative hallucination reduction vs. parameter-matched baseline.** Across hallucination benchmarks, ND-LoRA shows statistically significant improvements (HaluEval-Summarization, MemoTrap, TruthfulQA-MC2) while maintaining competitive Winogrande, NQ, and Wikitext BPB (lower is better) general-purpose capabilities. Baselines use higher LoRA ranks for parameter parity. * indicates $p < 0.05$.

Figure 3: **Reliability improves as neural diversity increases (lower $\mathcal{D}$).** Specifically, diversity ($\mathcal{D}$) is negatively correlated with HaluEval-Summarization performance (slope=-37.842, R²=0.237, p=0.002), consistent with $\mathbb{P}(H) \propto \mathcal{D}$ in Theorem 1.

Parameter efficiency is evident comparing ND-LoRA R16 ($P = 2$) against Qwen2.5-0.5B LoRA R32. Despite lower-rank adapters, ND-LoRA consistently outperforms the high-rank baseline on hallucination tasks, demonstrating that architectural diversity provides more value at equal capacity. This shows representational diversity, not parameter count, drives reliability gains in our experiment.

These findings establish neural diversity as a practical reliability mechanism. Consistent improvements across hallucination benchmarks with preserved general performance suggest ND-LoRA addresses fundamental reliability challenges rather than metric-specific optimization. Figure 3 demonstrates strong empirical correlation between neural diversity and performance, building intuition for the causal relationship established in subsection 5.1.

## 4.2 TASK-DEPENDENT OPTIMALITY

Further, the optimal diversity is task-dependent. Table 1 reveals striking task-dependent sensitivity patterns relative to the $P = 1$ baseline. Hallucination-focused tasks show the largest gains: HaluEval Summarization achieves +25.6% relative improvement at $P = 4$, HaluEval QA shows +23.4% at $P = 4$, and TruthfulQA MC2 shows +9.5% at $P = 2$ while MemoTrap benefits from higher diversity ($P = 8$, +8.8%). Notably, knowledge-intensive tasks like PopQA, TriviaQA and NQ show no improvement over baseline, which is expected as ND-LoRA does not add new sources of knowledge or try to improve recall of existing knowledge. This heterogeneity demonstrates that different tasks require different amounts of neural diversity to maximize reliability, with hallucination-focused tasks generally benefiting most from decorrelated representations.

| Task | $\Delta\mathcal{D}$ | $\Delta$ Score | SE | d | p-value | Sig. | N |
|---|---|---|---|---|---|---|---|
| HaluEval-Summ | 0.024 | -0.005 | 0.010 | 0.007 | $1.6 \times 10^{-5}$ | *** | 512 |
| MemoTrap v2 | 0.031 | -0.003 | 0.010 | 0.000 | $8.2 \times 10^{-5}$ | *** | 512 |
| TruthfulQA-MC2 | 0.025 | -0.007 | 0.009 | 0.018 | $3.3 \times 10^{-7}$ | *** | 512 |

Table 3: **Artificial corruption of neural diversity establishes statistical causality.** Perturbing neural diversity ($\Delta\mathcal{D} > 0$) causes accuracy drops across tasks with high statistical significance ($p < 0.001$) via paired t-tests with Fisher meta-analysis (N=4 sub-experiments × 128 samples each).

## 5 MECHANISTIC ANALYSIS

### 5.1 NEURAL DIVERSITY AS THE CAUSAL MEDIATOR

To establish causality beyond correlation, we perform artificial corruption interventions that directly manipulate cross-stream similarity.

**Experiment Design.** Starting with a pre-trained ND-LoRA $P = 4$ model, we inject a corruption hook at the RMSNorm layer that randomly substitutes the hidden state at randomly-chosen positions in a given stream from another stream, perturbing $\mathcal{D}$ while preserving activation magnitudes. We evaluate on a matched basis: each corrupted evaluation is paired with an uncorrupted baseline using identical samples and resampling indices. Across 4 sub-experiments with different random seeds, we collect $N = 128$ paired samples per task. This paired design maximizes statistical power by controlling sample-level variance, analyzed via paired t-tests with Fisher meta-analysis.

**Results.** Table 3 provides statistically robust evidence that neural diversity causally affects performance. All three tasks show highly significant accuracy drops ($p < 0.001$) when stream-level substitution perturbs diversity ($\Delta\mathcal{D} \approx 0.025$). While effect sizes are modest (0.3% to 0.7% score reduction) — likely because artificial stream substitution creates out-of-distribution corruption patterns — the statistical significance establishes causality beyond correlational association.

### 5.2 ABLATIONS

To isolate the contributions of ND-LoRA, we systematically ablate ND-LoRA components at fixed $P = 4$ streams. All variants maintain parameter parity through LoRA rank adjustments, enabling fair comparison. We measure inference-time diversity ($\mathcal{D}$) at the aggregation layer using evaluation samples, quantifying actual cross-stream correlation during inference.

Table 4 reveals a super-linear combination: independent LoRA (+2.9%) and Barlow Twins (+1.4%) sum to 4.3% but achieve 4.9% when combined (Stream LoRA-BT) — a 14% bonus. Targeting KVQ attention amplifies this further by 2.6× to +12.8% (ND-LoRA at fixed $P = 4$; maximum gains reach 14.6% when optimizing $P$ per-task, see Table 1). Neither component alone suffices: ParScale's near-complete collapse ($\mathcal{D} = 0.9990$) yields only +0.5%, while Stream LoRA without regularization achieves +2.9%, both less than a quarter of ND-LoRA's final impact. This establishes that both architectural capacity and explicit regularization are necessary for full impact.

Notably, ParScale's original work found prefix tuning superior to LoRA for mean loss (Table 6 in Chen et al. 2025). However, stream-aware LoRA is necessary for reducing tail probability: even with Barlow Twins, prefix tuning collapses streams ($\mathcal{D} = 0.9988$), while stream-aware LoRA enables decorrelation ($\mathcal{D} = 0.1530$). This illustrates how second-moment objectives require different architectural choices than first-moment objectives.

Counterintuitively, ND-LoRA achieves best performance (+12.8%) with *higher* $\mathcal{D} = 0.4112$ than Stream LoRA-BT's 0.1530. This reveals that strategic localization to representational bottlenecks matters more than maximizing global decorrelation: focusing LoRA and Barlow Twins on KVQ attention modules provides 2.6× amplification. This further reinforces how second-moment objectives differ architecturally from first-moment ones and, consistent with Table 1, that neural diversity is a task-dependent resource requiring strategic allocation to critical computational pathways.

| Variant | Streams | LoRA | Regul. | Target | $\mathcal{D}$ | $\overline{\Delta}\%$ Score | $\Delta$ Cost |
|---|---|---|---|---|---|---|---|
| Standard | 1 | Single | D | All | – | 0.0% | **1.0x / 1.0x** |
| ParScale | $P$ | Single | D | All | 0.9990 | +0.5% | 1.00003x / 1.1x |
| ParScale-BT | $P$ | Single | D + BT | All | 0.9988 | +1.4% | 1.00003x / 1.1x |
| Stream LoRA | $P$ | Stream | D | All | 0.3544 | +2.9% | 1.00003x / 1.1x |
| Stream LoRA-BT | $P$ | Stream | D + BT | All | **0.1530** | +4.9% | 1.00004x / 1.1x |
| ND-LoRA | $P$ | Stream | D + BT | KVQ | 0.4112 | **+12.8%** | 1.00004x / 1.1x |

Table 4: **Ablations reveal super-linear combination of impact.** Stream LoRA (+2.9%) and Barlow Twins (+1.4%) combine super-linearly (+4.9%), and focusing on KVQ attention amplifies to +12.8%. *LoRA*: single shared vs. $P$ stream-aware adapters. *Regularization*: Dropout vs. Barlow Twins. *Target*: All layers vs. KVQ attention only. $\mathcal{D}$: Neural Diversity Index (lower is better). $\overline{\Delta}\%$ *Score*: avg. change (hallucination benchmarks). Ablations shown at fixed $P = 4$ streams.

## 5.3 COMPUTATIONAL CONSIDERATIONS

Unlike $P$-model ensembles with $P\times$ pretraining cost, ND-LoRA achieves substantial reliability gains at negligible overhead ($1.00004\times$ pretraining, $1.1\times$ latency) given its single architecture. Parallelized 20M amortizes to $\sim 0.004\%$ of 1T-token pretraining, frozen backbone makes gradients nearly free, and ND-LoRA requires identical FLOPs to ParScale at inference. See subsection A.2.

## 5.4 PRACTICAL APPROXIMABILITY

While task-optimal $P_\star$ varies (Table 1), practitioners need not search exhaustively. Defaulting to $P = 4$ achieves 96% of oracle performance across all benchmarks. Additionally, a simple router (subsection A.7) achieves 97% by predicting $P$ from prompt statistics, revealing a retrieval-vs-verifiability tradeoff: question-dense prompts favor low $P$, while longer prompts favor higher $P$.

## 6 RELATED WORK

**Hallucination in Language Models.** Hallucinations represent a fundamental challenge in modern language models. Comprehensive surveys establish taxonomies that distinguish factuality vs. faithfulness (Huang et al., 2024; Tonmoy et al., 2024). Theoretical work proves hallucinations are mathematically inevitable in computable models under certain resource constraints (Xu et al., 2024; Kalai & Vempala, 2024), with smaller models exhibiting particular severity on factual benchmarks (Lin et al., 2021; Li et al., 2023a). Mechanistic investigations reveal hallucinations arise from internal representation failures (Yu et al., 2024), knowledge awareness limitations (Ferrando et al., 2025), and attention pattern anomalies.

Mitigation has predominantly targeted average performance. Retrieval augmentation (RAG) incorporates external knowledge for factual grounding (Niu et al., 2024). RLHF improves alignment (Bai et al., 2022), while constitutional AI enhances safety. Decoding methods use contrastive decoding (Li et al., 2023b) and classifier-free guidance (Sanchez et al., 2023). Critically, improving $\mathbb{E}[\text{error}]$ does not guarantee improvements to $\mathbb{P}(\text{hallucination})$, as tail events depend on variance and correlation structure, not just central tendency.

Second-moment approaches exist but lack theoretical grounding: self-consistency reduces hallucinations through diverse sampling (Wang et al., 2022) without formal tail-probability guarantees, while deep ensembles provide uncertainty estimates (Lakshminarayanan et al., 2017) but not hallucination-specific bounds. We provide the first formal tail bounds connecting neural diversity to hallucination probability as a second-moment problem.

**Deep Ensembles, Parallel Architectures & Inference-Time Scaling** Deep ensembles provide uncertainty estimates (Lakshminarayanan et al., 2017) with power-law scaling (Lobacheva et al., 2020) for calibration and OOD detection. LLM ensembles benefit from explicit diversity optimization (Tekin et al., 2024), while negative correlation learning demonstrates diversity must be actively encouraged (Liu & Yao, 1999). The "memory split advantage" shows ensembles of smaller models can outperform single large models at fixed parameter budgets. Optimal size theory reveals weighted voting exhibits diminishing returns due to correlation and overfitting (Bonab & Can, 2019), with

predictions stabilizing at 5–10 models (Hernández-Lobato et al., 2013). These approaches require multiple independent models, incurring $P\times$ training and inference costs.

Inference-time methods reduce hallucinations through diverse sampling and aggregation. Self-consistency uses majority voting over multiple generations (Wang et al., 2022). Confidence-based weighting uses intelligent aggregation (Taubenfeld et al., 2025), while contrastive decoding contrasts expert and amateur models (Li et al., 2023b). These approaches require multiple forward passes at inference time, whereas our training-time parallelism learns coordinated streams.

Self-ensembled parallel architectures like ParScale (Chen et al., 2025) break the multiplicative memory requirements of classical ensembles by using $P$ perturbed computational pathways within a single model. ParScale achieves $O(\log P)$ general capability gains, modeling parallel streams with correlation $\rho$ in scaling laws $L \propto (N \cdot P^{1/\alpha} \cdot [(P-1)\rho + 1]^{-1/\alpha})^{-\alpha}$. This targets mean loss for accuracy improvements, not hallucination probability. We directly build our demonstration on ParScale, extending their theoretical framework and implementation to tail-bound hallucinations.

**Theoretical Foundations.** Modern portfolio theory (Markowitz, 1952) provides the mathematical foundation for understanding correlation-based risk reduction, with diversification principles (Meucci, 2009) for ensemble variance analysis. Classical ensemble theory reduces mean error $\mathbb{E}[\text{loss}]$ via variance decomposition (Dietterich, 2000). PAC-Bayesian bounds connect diversity to minimax-optimal generalization (Ortega et al., 2022) and concentration inequalities showing correlation reduction tightens tail bounds (Alquier, 2024). We link these frameworks to modern neural networks to bound hallucination tail probabilities.

**Redundancy Reduction.** A rich history of diversification exists in self-supervised learning to avoid training collapse and in PEFT methods for efficient specialization. Self-supervised approaches like Barlow Twins (Zbontar et al., 2021) and VICReg (Bardes et al., 2022) use decorrelation to prevent dimensional collapse (Jing et al., 2022). PEFT methods like LoRA (Hu et al., 2022) and prefix-tuning (Li & Liang, 2021) enable model specialization under limited parameter budgets, with BatchEnsemble and LoRA-Ensemble achieving diversity through parameterization (Wen et al., 2020; Mühlematter et al., 2025). We adapt these methods for second-moment reliability guarantees.

## 7 DISCUSSION

At a time when the reliability of language models is becoming the critical barrier to real-world deployment, we (i) provide the first formal framework to tail-bound hallucinations in ensembled language models, demonstrating that neural diversity plays a critical role in reducing hallucinations; and, (ii) using this technique, achieve up to 25.6% (and 14.6% on average) reduction in hallucination rates at fixed parameter and data budgets at +0.004% pretraining cost. Neural diversity enables reliability gains without massive compute scaling.

By reframing hallucinations as a second-moment problem — controlled through variance and correlation rather than mean optimization — we open an under-explored research direction orthogonal to existing approaches. While RLHF and RAG target first-moment improvements (average performance), neural diversity targets tail probability through explicit decorrelation. This bridges portfolio theory to neural reliability, a connection previously unexplored. The gap between extensive first-moment research and nascent second-moment approaches (self-consistency, our work) suggests substantial opportunity for reliability-focused methods grounded in tail-probability theory.

Our small-scale demonstration and mechanistic analysis validates the theoretical framework; scaling to production models is straightforward given that continued training requires only +0.004% additional overhead and $P = 4$ captures 96.2% of oracle performance. The task-dependent optimal $P_\star$ in Table 1 reveals intriguing structure, suggesting deeper connections between task complexity, knowledge recall vs. precision and neural diversity worthy of theoretical characterization.

Our work opens two immediate research directions: (i) *Theoretical*: characterizing optimal $P_\star$ as a function of task properties — our U-shape theorem (Theorem 2) suggests information-theoretic approaches. (ii) *Practical*: combining neural diversity (this work) with inference-time scaling (Snell et al., 2024) for multiplicative reliability gains. Second-moment reliability is an essential frontier as LLMs become critical infrastructure in high-stakes domains.

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

## A    APPENDIX

### A.1    FULL PROOFS

#### A.1.1    PROOF OF LEMMA 1

*Proof.* We proceed in two steps. First, we show that the shared linear readout $A$ can at most distort cosines between representations by a factor of $\kappa^2$. Second, we convert bounded cosine alignment between the error vectors $\xi_i$ into a bound on average noise correlation.

**Step 1: local linear readout and cosine distortion.**    Let $\sigma_{\min}$ and $\sigma_{\max}$ denote the minimal and maximal singular values of $A$ and $\kappa = \sigma_{\max}/\sigma_{\min}$ its condition number. For any nonzero $x, y \in \mathbb{R}^d$, set $u = Ax$ and $v = Ay$. Then

$$\cos \angle(u, v) \;=\; \frac{\langle u, v \rangle}{\|u\|_2 \, \|v\|_2} \;=\; \frac{x^\top A^\top A y}{\|Ax\|_2 \, \|Ay\|_2}.$$

By the spectral bounds on $A^\top A$, we have

$$|x^\top A^\top A y| \;\leq\; \|A^\top A\|_{\mathrm{op}} \, |x^\top y| \;=\; \sigma_{\max}^2 \, |x^\top y|$$

and

$$\|Ax\|_2 \, \|Ay\|_2 \;\geq\; \sigma_{\min}^2 \, \|x\|_2 \, \|y\|_2.$$

Combining,

$$|\cos \angle(u, v)| \;\leq\; \frac{\sigma_{\max}^2}{\sigma_{\min}^2} \frac{|x^\top y|}{\|x\|_2 \, \|y\|_2} \;=\; \kappa^2 \, |\cos \angle(x, y)|.$$

Squaring both sides yields

$$\cos^2 \angle(u, v) \;\leq\; \kappa^4 \, \cos^2 \angle(x, y). \tag{10}$$

Apply this with $x = \tilde{z}_i(X)$ and $y = \tilde{z}_j(X)$. Under the local linear readout assumption we have $\xi_i(X) = A\tilde{z}_i(X)$ and $\xi_j(X) = A\tilde{z}_j(X)$, so $u = \xi_i(X)$ and $v = \xi_j(X)$. Thus, for every pair $(i, j)$ and every input $X$ with nonzero norms,

$$\cos^2 \angle\big(\xi_i(X), \xi_j(X)\big) \;\leq\; \kappa^4 \, \cos^2 \angle\big(\tilde{z}_i(X), \tilde{z}_j(X)\big).$$

Taking expectations over $X$ gives

$$\mathcal{D}_{\xi,ij}^2 \;\triangleq\; \mathbb{E}_X \big[ \cos^2 \angle(\xi_i(X), \xi_j(X)) \big] \;\leq\; \kappa^4 \, \mathbb{E}_X \big[ \cos^2 \angle(\tilde{z}_i(X), \tilde{z}_j(X)) \big] \;\triangleq\; \kappa^4 \, \mathcal{D}_{ij}^2,$$

where $\mathcal{D}_{ij}^2$ denotes the pairwise cosine diversity in representation space. Averaging over pairs and taking square roots yields

$$\mathcal{D}_\xi \;\triangleq\; \sqrt{\mathbb{E}_{i<j} \mathcal{D}_{\xi,ij}^2} \;\leq\; \kappa^2 \sqrt{\mathbb{E}_{i<j} \mathcal{D}_{ij}^2} \;=\; \kappa^2 \, \mathcal{D}. \tag{11}$$

**Step 2: from cosine alignment to average correlation.**    We now bound the average correlation $\bar{\rho}$ in terms of $\mathcal{D}_\xi$. Fix a pair $(i, j)$ and write

$$X \;\triangleq\; \langle \xi_i, \xi_j \rangle, \qquad B \;\triangleq\; \|\xi_i\|_2 \, \|\xi_j\|_2.$$

Whenever $B > 0$,

$$\cos \angle(\xi_i, \xi_j) \;=\; \frac{X}{B}, \qquad \mathcal{D}_{\xi,ij}^2 \;=\; \mathbb{E}\left[ \left( \frac{X}{B} \right)^2 \right].$$

Using Cauchy–Schwarz with $U = X/B$ and $V = B$, we obtain

$$\Sigma_{ij}^2 = \big( \mathbb{E}[X] \big)^2 = \big( \mathbb{E}[UV] \big)^2 \;\leq\; \mathbb{E}[U^2] \, \mathbb{E}[V^2] = \mathcal{D}_{\xi,ij}^2 \, \mathbb{E} \big[ \|\xi_i\|_2^2 \, \|\xi_j\|_2^2 \big].$$

Apply Cauchy–Schwarz again to the norms and use the kurtosis bound:

$$\mathbb{E} \big[ \|\xi_i\|_2^2 \, \|\xi_j\|_2^2 \big] \;\leq\; \sqrt{\mathbb{E}[\|\xi_i\|_2^4] \, \mathbb{E}[\|\xi_j\|_2^4]} \;\leq\; \sqrt{C_4 \sigma_i^4 \, C_4 \sigma_j^4} = C_4 \, \sigma_i^2 \sigma_j^2.$$

Combining,

$$\Sigma_{ij}^2 \;\leq\; C_4\,\mathcal{D}_{\xi,ij}^2\,\sigma_i^2\sigma_j^2, \qquad \rho_{ij}^2 = \frac{\Sigma_{ij}^2}{\sigma_i^2\sigma_j^2} \;\leq\; C_4\,\mathcal{D}_{\xi,ij}^2,$$

so

$$|\rho_{ij}| \;\leq\; \sqrt{C_4}\,\mathcal{D}_{\xi,ij}. \tag{12}$$

Finally, average over pairs and apply Cauchy–Schwarz in the index space:

$$|\bar{\rho}| = \left| \underset{i<j}{\mathbb{E}}[\rho_{ij}] \right| \;\leq\; \underset{i<j}{\mathbb{E}}[|\rho_{ij}|] \;\leq\; \sqrt{C_4}\,\underset{i<j}{\mathbb{E}}[\mathcal{D}_{\xi,ij}] \;\leq\; \sqrt{C_4}\,\sqrt{\underset{i<j}{\mathbb{E}}\,\mathcal{D}_{\xi,ij}^2} = \sqrt{C_4}\,\mathcal{D}_{\xi}.$$

Plugging equation 11 into this inequality gives

$$|\bar{\rho}| \;\leq\; \sqrt{C_4}\,\mathcal{D}_{\xi} \;\leq\; \sqrt{C_4}\,\kappa^2\,\mathcal{D} \;=\; C_*\,\mathcal{D},$$

with $C_* = \sqrt{C_4}\,\kappa^2$, as claimed. $\qquad\square$

### A.1.2 PROOF OF THEOREM 1

*Proof.* We work under the signal–noise model from the preliminaries. By Markowitz 1952,

$$\mathrm{Var}\left( \frac{1}{P} \sum_{i=0}^{P-1} X_i \right) = \bar{\sigma}^2 \left( \frac{1-\rho}{P} + \rho \right)$$

where $\bar{\sigma}^2 = \frac{1}{P}\sum_i \mathrm{Var}(X_i)$ is the average variance and $\rho = \mathbb{E}_{i\neq j}[\mathrm{Corr}(X_i, X_j)]$ is the average pairwise correlation.

From Lemma 1, we have

$$|\bar{\rho}| \;\leq\; C_*\,\mathcal{D}.$$

Substituting this into the variance expression yields

$$\mathrm{Var}(E_w) \leq \bar{\sigma}^2 \left( \frac{1 - C_*\,\mathcal{D}}{P} + C_*\,\mathcal{D} \right),$$

which is exactly the claimed variance bound in Theorem 1.

By construction, the hallucination event is

$$\mathrm{H}_\delta \;\triangleq\; \{E_w \geq \delta\}, \qquad \delta > 0,$$

and we have already noted that $\mathbb{E}[E_w] = 0$. Applying the one-sided Chebyshev inequality from the preliminaries to the random variable $E_w$ with mean 0 and variance $v = \mathrm{Var}(E_w)$ gives

$$\mathbb{P}(\mathrm{H}_\delta) \;=\; \mathbb{P}(E_w \geq \delta) \;\leq\; \frac{v}{v + \delta^2} \;=\; \frac{\mathrm{Var}(E_w)}{\mathrm{Var}(E_w) + \delta^2}.$$

Substituting $\mathrm{Var}(E_w)$ by its upper bound yields

$$\mathbb{P}(\mathrm{H}_\delta) \;\leq\; \frac{\bar{\sigma}^2 \left( \frac{1-C_*\,\mathcal{D}}{P} + C_*\,\mathcal{D} \right)}{\bar{\sigma}^2 \left( \frac{1-C_*\,\mathcal{D}}{P} + C_*\,\mathcal{D} \right) + \delta^2}.$$

Dividing both numerator and denominator by $\bar{\sigma}^2$ matches the bound stated in Theorem 1. $\qquad\square$

### A.1.3 PROOF OF THEOREM 2

*Proof.* Extend $P$ to a real variable with domain $P \geq 1$; the claim for integer $P$ follows by restriction.

Under uniform weights $w_i = 1/P$, the ensemble error variance can be written as

$$v(P) \triangleq \mathrm{Var}(E_w) = \bar{\sigma}^2 \left( \frac{1 - \bar{\rho}(P)}{P} + \bar{\rho}(P) \right),$$

with $\bar{\sigma}^2 > 0$ and

$$\bar{\rho}(P) = \rho_0 + \beta(P-1)^\gamma, \qquad \rho_0 \in [0,1), \; \beta > 0, \; \gamma > 0.$$

The bound from the main text is

$$\mathbb{P}(\mathrm{H}_\delta) \leq B(P) \triangleq \frac{v(P)}{v(P) + \delta^2}, \qquad \delta > 0.$$

**Step 1: Reduction to $v(P)$.** Define $\phi(x) \triangleq x/(x + \delta^2)$ for $x \geq 0$. Then

$$\phi'(x) = \frac{\delta^2}{(x + \delta^2)^2} > 0,$$

so $\phi$ is strictly increasing. Hence $B(P) = \phi(v(P))$ has the same extrema and monotonicity as $v(P)$. Since $\bar{\sigma}^2 > 0$, it suffices to analyze

$$f(P) \triangleq \frac{v(P)}{\bar{\sigma}^2} = \frac{1 - \bar{\rho}(P)}{P} + \bar{\rho}(P).$$

**Step 2: First derivative and unique critical point.** For $P > 1$,

$$\bar{\rho}'(P) = \beta\gamma(P - 1)^{\gamma - 1}.$$

A direct calculation gives

$$\begin{aligned}
f'(P) &= \frac{\mathrm{d}}{\mathrm{d}P}\left(\frac{1 - \bar{\rho}(P)}{P} + \bar{\rho}(P)\right) \\
&= \frac{\beta(P - 1)^\gamma(P\gamma + 1) + (\rho_0 - 1)}{P^2} \triangleq \frac{N(P)}{P^2}.
\end{aligned}$$

We study $N(P)$.

At $P = 1$ we have

$$N(1) = \beta \cdot 0^\gamma(\gamma + 1) + (\rho_0 - 1) = \rho_0 - 1 < 0.$$

Differentiating $N$ for $P > 1$ yields

$$N'(P) = \beta\gamma(\gamma + 1) \cdot P \cdot (P - 1)^{\gamma - 1}.$$

All factors on the right are strictly positive for $P > 1$, so $N'(P) > 0$ on $(1, \infty)$ and $N$ is strictly increasing. Moreover,

$$(P - 1)^\gamma(P\gamma + 1) \sim \gamma P^{\gamma + 1} \xrightarrow[P \to \infty]{} \infty,$$

so $N(P) \to +\infty$ as $P \to \infty$. By continuity and strict monotonicity, there exists a unique $P_\star > 1$ such that $N(P_\star) = 0$.

Because $P^2 > 0$ for all $P \geq 1$, the sign of $f'(P)$ matches that of $N(P)$:

$$f'(P) \begin{cases} < 0, & 1 < P < P_\star, \\ = 0, & P = P_\star, \\ > 0, & P > P_\star. \end{cases}$$

Thus $f$ (and hence $v$) is strictly decreasing on $(1, P_\star)$ and strictly increasing on $(P_\star, \infty)$; $P_\star$ is the unique global minimizer.

**Step 3: U-shape of the hallucination bound.** Since $B(P) = \phi(v(P))$ and $\phi$ is strictly increasing,

$$B'(P) = \phi'(v(P)) \cdot v'(P), \qquad \phi'(v(P)) > 0,$$

so $B$ inherits the same monotonicity: it is strictly decreasing on $(1, P_\star)$, strictly increasing on $(P_\star, \infty)$, and

$$B(P_\star) = \min_{P \geq 1} B(P).$$

Therefore the upper bound on $\mathbb{P}(H_\delta)$ is U-shaped in $P$ with a unique global minimum at $P_\star$, determined by the parameters $(\rho_0, \beta, \gamma)$ governing $\bar{\rho}(P)$. $\qquad\square$

## A.2 TRAINING COST AND LATENCY ANALYSIS

This appendix provides a complete analysis of computational costs for ND-LoRA and baseline variants. Three key insights enable negligible overhead: (1) fine-tuning on 20M tokens amortizes to less than 0.004% of 1T pretraining, (2) frozen backbone parameters make backward passes nearly free, and (3) inference uses identical FLOPs to ParScale via dynamic LoRA swapping per stream.

### A.2.1 COST MODEL

**Standard Fine-Tuning (P=1) Baseline.** Consider a standard LoRA fine-tuning setup with 495M backbone parameters frozen and 1.3M trainable adapter parameters. A typical training step consists of:

- **Forward pass**: $1.0\times$ computational cost through 495M parameters
- **Backward pass**: $2.0\times$ computational cost through 495M parameters (typical 2:1 backward:forward ratio)
- **Total baseline**: 3.0 cost units per training step

**ND-LoRA (P=4) Fine-Tuning.** With $P = 4$ parallel streams, ND-LoRA processes data through multiple independent pathways:

- **Forward pass**: $4.0\times$ cost (P parallel forward passes through full 495M model)
- **Backward pass**: $2.0 \times (1.3Y/495Y) \approx 0.005\times$ cost (gradients only propagate through 1.3M trainable parameters after aggregation)
- **Barlow Twins regularization**: $1.6\times$ cost (cross-correlation computation across P choose 2 streams and whitening)
- **Prefix/aggregator overhead**: $0.05\times$ cost (additional trainable components)
- **Total**: 5.655 cost units per training step

**Relative Training Cost.** The training cost of ND-LoRA relative to standard fine-tuning is:

$$\text{Relative cost} = \frac{5.655}{3.0} = 1.888\times \tag{13}$$

This is *substantially lower* than the naive estimate of $4\times$ because backward passes through frozen parameters are essentially free.

### A.2.2 AMORTIZATION OVER PRETRAINING

To contextualize fine-tuning costs, we amortize over typical pretraining budgets. Given:

- **Pretraining**: 1T tokens at $1.0\times$ cost = 1T token-equivalents
- **Fine-tuning**: 20M tokens at $1.888\times$ cost = 37.8M token-equivalents
- **Total**: $(1T + 37.8Y)/1T = 1.0000378 \approx 1.00004\times$

The amortized cost is **less than 0.004%** incremental overhead over the full training lifecycle.

### A.2.3 ALL VARIANTS DURING FINE-TUNING

Table 5 shows the complete cost breakdown for all ablation variants. The key differences are:

- **Shared vs. Stream-Aware LoRA**: Stream-aware adapters add $0.04\times$ prefix overhead
- **Barlow Twins**: Adds $1.6\times$ (full BT) or $0.1\times$ (ParScale-BT with simpler correlation)
- **All P=4 variants**: Incur $4.0\times$ forward pass cost but only $0.005\times$ backward cost

### A.2.4 INFERENCE LATENCY

At inference, all $P > 1$ variants exhibit $1.1\times$ **latency** relative to standard models:

- **Parameter parity**: All variants maintain identical total parameter counts by adjusting LoRA rank
- **Parallel processing**: P streams process in parallel; latency dominated by slowest stream + aggregation

| Variant | Forward | Backward | BT | Other | Total | Relative |
|---|---|---|---|---|---|---|
| Standard | 1.0 | 2.0 | 0.0 | 0.0 | 3.0 | 1.000× |
| ParScale | 4.0 | 0.005 | 0.0 | 0.01 | 4.015 | 1.337× |
| ParScale-BT | 4.0 | 0.005 | 0.1 | 0.01 | 4.155 | 1.384× |
| Indep. LoRA | 4.0 | 0.005 | 0.0 | 0.05 | 4.055 | 1.352× |
| ND-LoRA | 4.0 | 0.005 | 1.6 | 0.05 | 5.655 | 1.885× |

Table 5: Fine-tuning cost breakdown (20M tokens). *Forward*: P parallel passes through 495M backbone. *Backward*: single pass through 1.3M trainable parameters. *BT*: Barlow Twins correlation computation. *Other*: prefix/aggregator overhead. *Relative*: cost relative to 3.0× standard baseline.

- **Dynamic loading**: Different LoRA adapters are dynamically loaded per stream without duplication
- **Aggregation overhead**: Lightweight MLP aggregator adds ∼10% latency

The 1.1× factor is consistent across ParScale, ParScale-BT, Indep. LoRA, and ND-LoRA because inference does not involve Barlow Twins regularization and all parameter operations are equivalent.

### A.2.5 SUMMARY

- **Training overhead**: 1.89× (not 4×) due to free backward passes through frozen parameters
- **Amortized cost**: $\leq 0.004\%$ when amortized over 1T-token pretraining
- **Inference latency**: 1.1× across all $P \geq 1$ variants with parameter matching
- **Practical impact**: Negligible computational overhead for 25.6% hallucination reduction

### A.3 LoRA MODULE ABLATIONS

To understand which components of ND-LoRA contribute most to hallucination reduction, we perform targeted ablations by removing LoRA adapters from specific module types. We compare the full ND-LoRA baseline against two variants:

- *No MLP*: removing LoRA from MLP projections (gate_proj, up_proj, down_proj) while keeping attention LoRA
- *No Attention*: removing LoRA from attention projections (q_proj, k_proj, v_proj, o_proj) while keeping MLP LoRA

| Task | $\Delta$% No MLP | $\Delta$% No Attention |
|---|---|---|
| HaluEval Dialog | -1.7% | -0.6% |
| HaluEval QA | +16.8% | -1.8% |
| HaluEval Summarization | -5.3% | -27.0% |
| MemoTrap v2 | +2.5% | +0.9% |
| NQ (8-shot) | +11.7% | -1.7% |
| PopQA | -0.8% | -0.8% |
| TriviaQA (8-shot) | -5.0% | -6.9% |
| TruthfulQA MC1 | +3.1% | +2.4% |
| TruthfulQA MC2 | +0.2% | +1.4% |

Table 6: LoRA module ablation results (relative percentage changes from baseline). Evaluations performed on N=1024 samples per task.

### A.4 USE OF LARGE LANGUAGE MODELS

Large language models were used as a compilation tool to assist with writing and organizing sections of this paper, including literature review synthesis, section structuring, LaTeX formatting, and co-generation of experimental code. All technical content, experimental design, theoretical contributions, and scientific claims are the authors' original work. The models served primarily to improve clarity, organization, and implementation of our ideas rather than generate novel scientific insights.

## A.5 EXPERIMENTAL SETUP

**Model and Architecture.** We use Qwen2.5-0.5B (896 hidden dimensions, 24 layers) with ND-LoRA across $P \in \{1, 2, 4, 8\}$ parallel streams applied to QKV self-attention modules and a design layer of 20 for de-correlation loss. Each stream uses independent rank-16 LoRA adapters and 48 prefix tokens, totaling 5-20M trainable parameters with 495M backbone frozen. Baseline methods use higher-rank LoRA (R32-R128) for parameter matching.

**Training Protocol.** Models train on 20M tokens from The Pile (8 random shards, fixed seeds). We use 1024-token sequences, AdamW optimization (peak lr 3e-4, cosine decay, 2% warmup), batch size 64, bfloat16 precision. Training completes in ~5K steps (~30 min. on A100).

**Evaluation Benchmarks.** We evaluate across: (1) *Hallucination-sensitive*: TruthfulQA (Lin et al., 2021), HaluEval (Li et al., 2023a), MemoTrap (McKenzie et al., 2023); (2) *Knowledge-intensive*: Natural Questions (Kwiatkowski et al., 2019), TriviaQA (Joshi et al., 2017), PopQA (Mallen et al., 2023); (3) *General capability*: Wikitext BPB (Merity et al., 2017), Winogrande (Sakaguchi et al., 2020). This tests if neural diversity improves reliability without sacrificing general performance.

**Neural Diversity Measurement.** We compute $\mathcal{D}$ at the final RMSNorm layer by first whitening representations per feature dimension across batch and sequence positions (zero mean, unit variance), then computing pairwise cosine similarity between streams. This is equivalent to the Barlow Twins cross-correlation formulation (Eq. 2 in Zbontar et al. (2021)) when features are whitened.

**Statistical Methodology.** We evaluate significance using McNemar's test for binary classification tasks and two-tailed bootstrap tests with 10,000 samples for other tasks. Improvements marked with * are significant at $p < 0.05$.

## A.6 COMPLETE BENCHMARK RESULTS

Tables 7–9 provide comprehensive results across $P \in \{1, 2, 4, 8\}$ configurations with parameter-matched $P = 1$ baselines. This complete view demonstrates the thoroughness of our evaluation and enables independent verification of claims in the main text.

| Evaluation | Qwen LoRA | ParScale | ND-LoRA |
|---|---|---|---|
| HE Dialog | 0.458 | 0.453 | **0.513** |
| HE QA | 0.365 | 0.337 | **0.406** |
| HE Summ | 0.400 | 0.439 | **0.481** |
| MemoTrap | 0.634 | 0.638 | **0.666** |
| NQ-8 | **0.065** | 0.059 | 0.055 |
| TQA-8 | **0.188** | 0.185 | 0.160 |
| TF-MC1 | 0.251 | 0.259 | **0.269** |
| TF-MC2 | 0.403 | 0.412 | **0.442** |
| NQ-swap | **0.550** | 0.546 | 0.528 |
| PopQA | **0.111** | 0.109 | 0.101 |
| Wikitext BPB | **0.775** | 0.797 | 0.797 |
| Winogrande | 0.572 | 0.564 | **0.574** |

Table 7: Benchmark results for $P = 2$ (Qwen R32) parameter-matched models.

## A.7 AN INTERPRETABLE ROUTER FOR OPTIMAL NUMBER OF STREAMS

To demonstrate that the task-optimal $P_\star$ patterns in Table 1 reflect real structure rather than arbitrary variation, we train a simple interpretable router that predicts optimal $P_\star$ from prompt features alone. While more complex routers could improve performance, we prioritize simplicity and interpretability to understand the underlying structure.

We fit a simple regression on two features, trained on just 10 samples per task with oracle $P$ labels:

$$\hat{P} = \text{clip}(0.196 \log W - 2.283Q + 3.321) \tag{14}$$

| Evaluation | Qwen LoRA | ParScale | ND-LoRA |
|---|---|---|---|
| HE Dialog | 0.464 | 0.459 | **0.516** |
| HE QA | 0.341 | 0.322 | **0.451** |
| HE Summ | 0.394 | 0.409 | **0.502** |
| MemoTrap | 0.629 | 0.634 | **0.635** |
| NQ-8 | **0.065** | 0.061 | 0.059 |
| TQA-8 | **0.191** | 0.185 | 0.172 |
| TF-MC1 | 0.245 | 0.253 | **0.262** |
| TF-MC2 | 0.399 | 0.413 | **0.416** |
| NQ-swap | **0.554** | 0.542 | 0.535 |
| PopQA | **0.110** | **0.110** | 0.106 |
| Wikitext BPB | **0.778** | 0.793 | 0.795 |
| Winogrande | 0.564 | 0.573 | **0.577** |

Table 8: Benchmark results for $P = 4$ (Qwen R64) parameter-matched models.

| Evaluation | Qwen LoRA | ParScale | ND-LoRA |
|---|---|---|---|
| HE Dialog | 0.460 | 0.465 | **0.475** |
| HE QA | 0.344 | 0.335 | **0.370** |
| HE Summ | 0.379 | 0.416 | **0.450** |
| MemoTrap | 0.630 | 0.639 | **0.689** |
| NQ-8 | **0.066** | 0.063 | 0.059 |
| TQA-8 | **0.192** | 0.182 | 0.171 |
| TF-MC1 | 0.251 | 0.256 | **0.259** |
| TF-MC2 | 0.407 | 0.414 | **0.424** |
| NQ-swap | 0.551 | 0.540 | **0.554** |
| PopQA | **0.110** | 0.109 | 0.103 |
| Wikitext BPB | **0.778** | 0.779 | 0.784 |
| Winogrande | 0.569 | **0.577** | 0.568 |

Table 9: Benchmark results for $P = 8$ (Qwen R128) parameter-matched models.

where $Q$ is the ratio of interrogative to declarative sentences, $W$ measures prompt length in words, and clip$(\cdot)$ snaps predictions to the nearest valid $P \in \{1, 2, 4, 8\}$. This two-feature router achieves 96.8% of oracle performance on held-out samples averaged across all tasks.

The learned coefficients reveal an interpretable trade-off between *knowledge retrieval* and *verifiability*. The negative weight on interrogative sentence ratio indicates that question-dense prompts — where success depends on precise recall of stored knowledge — benefit from lower $P$ values that maximize focus from a single stream. Conversely, the positive weight on word count reflects that longer prompts — where success depends on cross-checking claims against provided context — require higher $P$ for diverse verification across streams. More broadly, tasks prioritizing retrieval favor low diversity, while tasks prioritizing verifiability favor high diversity.

### A.8 LoRA Hyperparameters as Potential Confounds

A natural concern is whether ND-LoRA's improvements stem from LoRA hyperparameter choices rather than neural diversity *per se*. We consider three potential confounds: (i) *expressivity*: $P$ parallel rank-$R$ adapters yield $P \times R$ total parameters, so improvements might reflect capacity rather than diversity; (ii) *alpha scaling*: different $\alpha/r$ ratios affect update magnitudes and could change which solutions are reachable; and (iii) *optimization dynamics*: higher-rank adapters might converge to different basins.

**Expressivity.** This confound is addressed by parameter matching in the main text (Table 2): baselines use higher-rank single LoRA (R32–R128) to match ND-LoRA's total parameter count, yet ND-LoRA still outperforms on hallucination benchmarks.

**Alpha scaling.** We conducted a sensitivity analysis varying single-LoRA rank from $R16$ to $R128$ under two alpha strategies: constant scaling ($\alpha/r = 2$) and constant alpha ($\alpha = 32$). Results in Table 10 show that constant-scaling single-LoRA is not a suitable baseline for two reasons. First, the only monotonic trend observed is *degradation* of general capabilities: Wikitext perplexity increases from 0.776 to 0.795 bits per byte (+2.4%), TriviaQA-8 drops from 19% to 17% (-11%), and NQ-8 drops from 7% to 5% (-29%) as rank increases from $R16$ to $R128$. Second, hallucination benchmark performance is unstable across this $8\times$ rank variation: while some pairwise differences are statistically significant, they're unstable across both rank and benchmarks (e.g. HE-Dialog vs. HE-QA within R64). We therefore use fixed $\alpha = 32$ baselines, which provide stable reference points without the capability degradation observed under constant scaling. Importantly, ND-LoRA remains statistically significantly better than both baseline types — all winners stay winners — and using constant-scaling baselines would in fact create additional ND-LoRA wins (e.g. Wikitext BPB, NQ-8 and Winogrande $P = 8$).

| Metric | R16 | R32 | R64 | R128 |
|---|---|---|---|---|
| HE Dialog | 0.46±0.01 | 0.46±0.01 | 0.49±0.01 | 0.45±0.01 |
| HE QA | 0.37±0.01 | 0.37±0.01 | 0.34±0.01 | 0.36±0.01 |
| HE Summ | 0.41±0.01 | 0.46±0.01 | 0.48±0.01 | 0.41±0.01 |
| MemoTrap | 0.64±0.03 | 0.63±0.03 | 0.63±0.03 | 0.64±0.03 |
| TF-MC1 | 0.25±0.03 | 0.25±0.03 | 0.24±0.03 | 0.24±0.03 |
| TF-MC2 | 0.41±0.03 | 0.40±0.03 | 0.39±0.03 | 0.40±0.03 |
| NQ-8 | 0.07±0.01 | 0.06±0.01 | 0.06±0.01 | 0.05±0.01 |
| NQ-swap | 0.55±0.01 | 0.55±0.01 | 0.55±0.01 | 0.54±0.01 |
| PopQA | 0.11±0.01 | 0.11±0.01 | 0.11±0.01 | 0.11±0.01 |
| TQA-8 | 0.19±0.01 | 0.18±0.01 | 0.18±0.01 | 0.17±0.01 |
| Wikitext BPB | 0.776 | 0.781 | 0.790 | 0.795 |
| Winogrande | 0.56±0.03 | 0.57±0.03 | 0.58±0.03 | 0.56±0.03 |
| $\alpha/r$ | 2.00 | 2.00 | 2.00 | 2.00 |

Table 10: Constant scaling $\alpha/r = 2$: $\alpha$ varies with rank. Hallucination metrics are noisy but many general-capability metrics degrade monotonically (e.g. Wikitext BPB 0.776 → 0.795, NQ-8 0.07 → 0.05), making this an unsuitable baseline.

| Metric | R16 | R32 | R64 | R128 |
|---|---|---|---|---|
| HE Dialog | 0.46±0.01 | 0.46±0.01 | 0.46±0.01 | 0.46±0.01 |
| HE QA | 0.37±0.01 | 0.37±0.01 | 0.34±0.01 | 0.34±0.01 |
| HE Summ | 0.41±0.01 | 0.40±0.01 | 0.39±0.01 | 0.38±0.01 |
| MemoTrap | 0.64±0.03 | 0.63±0.03 | 0.63±0.03 | 0.63±0.03 |
| TF-MC1 | 0.25±0.03 | 0.25±0.03 | 0.24±0.03 | 0.25±0.03 |
| TF-MC2 | 0.41±0.03 | 0.40±0.03 | 0.40±0.03 | 0.41±0.03 |
| NQ-8 | 0.07±0.01 | 0.07±0.01 | 0.06±0.01 | 0.07±0.01 |
| NQ-swap | 0.55±0.01 | 0.55±0.01 | 0.55±0.01 | 0.55±0.01 |
| PopQA | 0.11±0.01 | 0.11±0.01 | 0.11±0.01 | 0.11±0.01 |
| TQA-8 | 0.19±0.01 | 0.19±0.01 | 0.19±0.01 | 0.19±0.01 |
| Wikitext BPB | 0.776 | 0.775 | 0.778 | 0.778 |
| Winogrande | 0.56±0.03 | 0.57±0.03 | 0.56±0.03 | 0.57±0.03 |
| $\alpha/r$ | 2.00 | 1.00 | 0.50 | 0.25 |

Table 11: Constant $\alpha = 32$: scaling varies with rank. Most metrics are stable alongside general capabilities, helping rule out expressivity and optimization dynamics as confounds.

**Optimization dynamics.** Under fixed alpha ($\alpha = 32$), general capabilities remain stable across $8\times$ rank variation: Wikitext BPB is flat (0.775–0.778) and Winogrande accuracy is statistically indistinguishable (0.56–0.57) across R16–R128 (Table 11). If optimization dynamics differed meaningfully across rank (e.g. higher-rank adapters converging to different loss basins) we would expect divergence on these general capability metrics. The observed stability indicates that fixed-alpha configurations converge to similar solutions regardless of rank, ruling out optimization dynamics as a confound for ND-LoRA's hallucination improvements.

