# OpenReview forum: "Neural Diversity Regularizes Hallucinations in Language Models"
_ICLR.cc/2026/Conference — Submitted to ICLR 2026_

### Official Review · Reviewer_pUug · 2025-10-26

**Soundness:** 1
**Presentation:** 1
**Contribution:** 2
**Rating:** 2
**Confidence:** 4

**Summary:**

The paper explores a connection between higher diversity and hallucination rate in small language models.

**Strengths:**

- The paper investigates a relevant research question: how to reduce the rate of hallucinations of a small language model.
- The paper presents a broad set of experiments on several evaluation benchmarks.

**Weaknesses:**

The clarity and rigor of the paper could be greatly improved.
- The manuscript is often not self-contained and some of the terminology is not defined before being mentioned. For instance, the abstract mentions “neural diversity”, without describing what it refers to, Figure 1 shows metrics that are not well-defined at this point, line 52 makes a statement about “parallel streams” without properly introducing what they are etc.
- Several claims in the paper are not sufficiently precise. For instance, the contributions in lines 74-76 state “17.9% reduction at 1.12x cost” (reduction in what? what cost?). Moreover, the preliminaries introduce some notation, without properly describing what it refers to. What are the $m_i$ streams? What is the hallucination event and why can it be defined like in line 99?
- Section 2.2 is rather difficult to read. It is not clear what the main takeaways are, and why theorem 1 is significant. It references results that are not present in the main text (a certain Corollary in line 144) and notation that is never defined (e.g. $\overline{\kappa}$).

The paper claims to establish a causal link between diversity and the rate of hallucinations (line 75). However, it is unclear what evidence points to that.

**Minor issues**

- incorrect references in the latex document e.g. lines 315, 583, 600 etc
- figures 1 and 2 are never references in the main text

**Questions:**

- In lines 100-101 it is stated “The margins [...] represent stream confidence—positive indicates correctness, negative indicates error.” Generally, for prediction tasks, confidence and accuracy are completely independent metrics (a predictor can be correct or not while being confident – or not). Why is it the case that here high confidence is correlated with correctness?

---

> ### Author Response · Authors · 2025-11-17
>
> Dear Reviewer pUug,
>
> Thank you for your thorough review. Your focus on clarity and self-containment identified exactly where we needed to improve—we've substantially revised the manuscript to be more rigorous and self-contained. We're encouraged that the core concerns are addressable through better exposition.
>
> ---
>
> (answering jointly for brevity)
> > **W1:** The manuscript is often not self-contained and some of the terminology is not defined before being mentioned. For instance, the abstract mentions "neural diversity", without describing what it refers to, Figure 1 shows metrics that are not well-defined at this point, line 52 makes a statement about "parallel streams" without properly introducing what they are etc.
>
> > **W2:** Several claims in the paper are not sufficiently precise. For instance, the contributions in lines 74-76 state "17.9% reduction at 1.12x cost" (reduction in what? what cost?). Moreover, the preliminaries introduce some notation, without properly describing what it refers to. What are the streams?
>
> You're right — the original submission had too many forward references and insufficiently characterized claims. We've systematically addressed these issues: all forward references, insufficient precision, missing notation, and missing references/corollaries (including those you've identified but also others).
>
> ---
>
> > **W2 (definition):** "What is the hallucination event and why can it be defined like in line 99?"
>
> > **W3:** "Section 2.2 is rather difficult to read. It is not clear what the main takeaways are, and why theorem 1 is significant. It references results that are not present in the main text (a certain Corollary in line 144) and notation that is never defined."
>
> > **Q1:** In lines 100-101 it is stated "The margins [...] represent stream confidence—positive indicates correctness, negative indicates error." Generally, for prediction tasks, confidence and accuracy are completely independent metrics (a predictor can be correct or not while being confident – or not). Why is it the case that here high confidence is correlated with correctness?
>
> Fair point – our original formulation was confusing. Rather than go through gymnastics to defend a poor setup, we describe our revised formalization. Our analysis proceeds in three basic steps:
>
> (1) **Framework:** Each of P parallel streams outputs Y_i = μ + ε_i (oracle truth + noise). Hallucination occurs when averaged output (1/P) Σ Y_i ≤ 0, i.e. when aggregated noise overwhelms the signal.
>
> (2) **Diversity bounds:** By decorrelating noise ε_i across streams, we derive tail bounds on P(H) as a function of signal strength μ, noise variance σ², inter-stream correlation ρ, and stream count P. When perfectly decorrelated, we show P(H) ~ 1/P.
>
> (3) **Optimal parallelism:** When ρ increases with P (e.g., from optimizer constraints), we prove there exists an optimal P_* that minimizes hallucination probability.
>
> Empirically, this framework works — our theoretical model explains 96.2% of empirical reliability variation (R²=0.962), showing that hallucinations can in fact be tractably analyzed this way.
>
> In addition, each key result now has a dedicated paragraph explaining (1) what it shows, (2) why it matters, and (3) how it connects to empirical results. All notational issues have been fixed.
>
> ---
>
> > **W4:** "The paper claims to establish a causal link between diversity and the rate of hallucinations (line 75). However, it is unclear what evidence points to that."
>
> We've added four types of evidence to establish the mechanistic link:
>
> - **(new) Table 3 (Intervention)**: Artificially perturbing correlation and measuring downstream hallucination effects (p < 0.001).
> - **(new) Figure 3 (Correlation)**: Sample-level correlation between spectral diversity D_spec and hallucination rate (R²=0.237, p=0.002).
> - **(augmented) Table 4 (Ablations)**: D_spec measurements across all ablation variants showing mechanistic relationship.
> - **(augmented) Figure 1 (Explanatory)**: Explanatory ability of Theorems 1 & 2 for observed reliability gains across parallel configurations (R^2 = 0.962)
>
> Combined with theoretical grounding, these establish a strong mechanistic argument that neural diversity mediates hallucination.
>
> ---
>
> > **Minor issues:** "Incorrect references in the latex document e.g. lines 315, 583, 600 etc; figures 1 and 2 are never referenced in the main text"
>
> Fixed. All LaTeX references corrected, Figures 1-2 now referenced in main text (lines 215, 220, 253).
>
> ---
>
> We're grateful for your detailed feedback, which significantly improved the paper's clarity and rigor. We would welcome any additional feedback – our goal is a clear, honest, and impactful paper.

---

> ### Author Response · Authors · 2025-12-01
>
> Dear Reviewer pUug,
>
> We understand you can't respond given the circumstances, but wanted to acknowledge your feedback and say thank you. Your focus on clarity and self-containment identified exactly where we needed to improve—we've substantially revised the manuscript to be more rigorous and accessible.
>
> We've uploaded [another revision](https://openreview.net/pdf?id=qQoLZ25tIs); in total, your concerns addressed:
> - **W3 + Q1 (Theory section):** Section 2 completely rewritten with (i) clearer structure, motivation and interpretation/takeaways and (ii) reworked under a signal-noise model, Lipschitz decoding and Lemma 1 exploiting shared-backbone structure to concretely link LM hallucination to portfolio theory under real-world assumptions. Our theoretical framework is validated by R²=0.962 (augmented Figure 1) against empirical reliability gains.
> - **W4 (Causality):** We add three new lines of evidence: (i) Interventions perturbing diversity cause accuracy drops (Table 3, p<0.001 all tasks); (ii) +0.1% neural correlation ↔ +3.8% hallucination (Figure 3, p=0.002); (iii) Ablations mechanistically track diversity-hallucination relationship (Table 5).
> - **W1 (Self-containment):** All terms (neural diversity, parallel streams) now defined before use; all figures referenced in main text
> - **W2 (Precision):** Claims now fully specified (e.g., "25.6% hallucination reduction" → "up to 25.6% (14.6% average) hallucination reduction at +0.004% pretraining cost")
> - **Minor issues:** All LaTeX errors fixed, all figures referenced
>
> We hope the revision addresses your concerns.
>
> With appreciation,
> The Authors

---

### Official Review · Reviewer_EyTf · 2025-10-31

**Soundness:** 3
**Presentation:** 3
**Contribution:** 3
**Rating:** 4
**Confidence:** 4

**Summary:**

The paper argues for 'neural diversity', i.e., diversity in model architecture, as an important component to reduce hallucinations in parallel scaling of small language models. It builds on the work of ParScale, a framework for model scaling by running parallel streams of the model, and argues that these parallel streams should also produce diverse features to reduce hallucinations. The paper supports its arguments, first using theoretical bounds and then with empirical experiments for the same.

**Strengths:**

1. The inclusion of 'ND-LoRA' by the paper to increase diversity clearly reduces hallucinations, without a significant increase in computation cost. In other words, the biggest strength of the paper is that the technique proposed works. While additional experiments are always appreciated, I believe the current set of experiments are robust enough to suggest that the technique can be expected to work in other settings.

2. Ablation studies and experiments across multiple datasets show robust benefits of the technique proposed by the paper.

3. The theoretical bounds and discussions are solid (although not novel, discussed further in Weaknesses), and thus the work is grounded in strong foundations.

**Weaknesses:**

Comment on Related Work: The paper needs a better treatment of related work. Many weaknesses discussed below will refer back to this particular lack of appropriate discussion of related works in the paper. There is a section on Related Works towards the end of the paper, but it does not do justice to the rest of the discussion in the paper.

1. I fail to separate the novelty of the theoretical analysis provided in the paper from those that already exist in the ensemble literature. From the general discussion of diversity (see [1], [2]) to an 'optimal ensemble size' (see [3], [4]), and even specific claims like 'signal-to-noise improves by root(P)' (see [4]), have an extensive history in ensemble literature. The references provided are only examples, there is far more literature on ensemble theory. While the discussion in the paper is provided in the context of 'hallucinations', which could have been a novelty, the theoretical analysis simply reduces the hallucination aspect to margins at the very start and thus loses any context-specific addition that it can provide to the literature.
The paper briefly acknowledges this in Related Works, based on which, it seems that the main contribution is simply that existing literature assumes separate models, but the paper shows benefits for the same architecture. Firstly, separate 'models' and same 'architecture' are two different things, and are not comparable. Secondly, the paper never makes use of 'same architecture' in their theoretical analysis, which only assumes different features (hence, actually assumes different models, same as in the literature). And finally, none of this is reflected in the rest of the paper, which is framed as if the theoretical discussions are contributions of the paper.

2. The connections with ParScale and existing discussions of diversity in the original ParScale paper should be acknowledged and also further discussed in this paper. For instance, ParScale paper claims that using learnable prefixes provides 'diverse enough' feature outputs, which is in contrast with the claim made in the introduction of this paper that naive scaling can degrade reliability (this claim is also in contrast with the empirical results provided by this paper itself in the ablation study). To my understanding, the paper needs to readjust its claim to say that explicitly incorporating diversity can 'further' improve reliability (reduce hallucinations), or add a discussion on why their claim stands despite existing literature and their own results.
On a similar line, the inclusion of LoRA is surprising to me. Original ParScale paper experiments with LoRA alongside the learnable prefix, and concludes that the prefix is a better choice. Why was then LoRA used in this paper? Arguably, LoRA has a higher rank in this paper, compared to the original ParScale paper, which might be the difference (again, this is based on just my observations from the two papers, and no such discussion of how their choices build on previous work exists in this paper). A proper motivation behind the choices made is needed (a simple ablation study is not enough).

3. There is a missing corollary (line 144) in the theoretical analysis, and a missing figure (line 315) in the key results. The missing corollary is not too important in context, since I don't believe the theoretical analysis is a novelty. However, the missing figure is quite important, as it supposedly contains the connections between theoretical bounds and empirical results.

4. I don't like the claim in the abstract, 'ND-LoRA achieves 17.9% hallucination reduction with 1.12× training cost'. The 17.9% improvement occurs only in one subset of one dataset, and the rest of the results are in the range of 1-8% improvements. To be clear, these improvements are still good. I understand the choice made by the authors to showcase their strongest result, but still, the language is misleading. I would recommend (although I'm still not a fan of this) something like 'ND-LoRA achieves up to 17.9% hallucination reduction in some dataset with 1.12× training cost'.

References

[1] Dietterich, Thomas G. "Ensemble methods in machine learning." International workshop on multiple classifier systems. Berlin, Heidelberg: Springer Berlin Heidelberg, 2000.

[2] Kuncheva, Ludmila I. Combining pattern classifiers: methods and algorithms. John Wiley & Sons, 2014.

[3] Bonab, Hamed, and Fazli Can. "Less is more: A comprehensive framework for the number of components of ensemble classifiers." IEEE Transactions on neural networks and learning systems 30.9 (2019): 2735-2745.

[4] Hernández-Lobato, Daniel, Gonzalo Martínez-Muñoz, and Alberto Suárez. "How large should ensembles of classifiers be?." Pattern Recognition 46.5 (2013): 1323-1336.

**Questions:**

1. Is there work that shows 'naive scaling' can degrade reliability (as claimed in the introduction)? It seems, based on Table 3, that even naive parallel scaling increases the score (no doubt that increasing diversity works even better, but is there a reasoning behind the base claim itself?).

2. Are there contributions in the theoretical foundations that are novel? If so, I recommend highlighting them explicitly, by contrasting with what has been done before and what the paper adds to the literature.

3. Please provide the missing information, as highlighted above in the weaknesses.

---

> ### Author Response · Authors · 2025-11-17
>
> Dear Reviewer EyTf,
>
> Thank you for the expert and thorough review. Your questions about novelty were exactly the push we needed to sharpen our contributions - we have substantially revised the Introduction, Section 2 and Related Works accordingly. We were surprised to find these appear to be the first hallucination tail bounds for ensembled LMs, and encouraged that you found it simply works.
>
> ---
>
> (answering jointly for brevity)
> > **W1 + Q2:** "I fail to separate the novelty of the theoretical analysis [...] from those that already exist in the ensemble literature. From the general discussion of diversity to an 'optimal ensemble size', and even specific claims like 'signal-to-noise improves by root(P)' [...]"
>
> You're right — Section 2 was poorly written. We've substantially revised it to (a) build upon ensemble/portfolio theory and (b) contrast our contributions directly in both Related Works and Section 2:
>
> Our contributions beyond established foundations (diversity improves generalization, √P scaling for independent errors, diminishing returns/overfitting, mean-variance analysis):
> - **First hallucination tail bound in ensembled LMs**: To our knowledge, we provide the first formal tail bounds on P(hallucination) for ensembled LMs, reframing it as a second-moment reliability problem.
> - **Portfolio theory linkage (Theorem 1)**: We adapt mean-variance portfolio theory to neural hallucinations, proving P(H) ~ 1/P for P perfectly-decorrelated representations.
> - **Non-monotonicity (Theorem 2)**: We prove and validate (Figure 1) that excessive parallelism can degrade diversity (and thus reliability) in ensembled parallel architectures.
> - **Quantitative validation (R²=0.962)**: Our theory works — explaining 96.2% of empirical reliability variation with a simple theoretical model.
>
> The key distinction: classical ensembles optimize E[loss]; we optimize P(H) (tail events)—same variance machinery, different objective.
>
> To be clear: we don't claim fundamental advances in ensemble or portfolio theory itself. Rather, we provide the first formal treatment of hallucination tail-risk in ensembled LMs using established variance-reduction machinery for a different objective (tail events vs. mean error).
>
> ---
>
> > **W2:** "The connections with ParScale [...] should be acknowledged and discussed. For instance, ParScale claims that learnable prefixes provide 'diverse enough' feature outputs, which contrasts with the claim [...] that naive scaling can degrade reliability. [...] The inclusion of LoRA is surprising to me. Original ParScale paper [...] concludes that the prefix is a better choice. Why was then LoRA used in this paper?"
>
> > **Q1:** "Is there work that shows 'naive scaling' can degrade reliability (as claimed in the introduction)? [...] based on Table 3, even naive parallel scaling increases the score."
>
> This is very fair. We've expanded the ParScale discussion (revised Section 6), tightened our naive scaling language, and explained the LoRA vs. prefix choice.
>
> **re: naive scaling claims:** Confusing claim — revised. Table 4 shows naive parallelism yields some improvement (but statistically insignificant). Our "degrading reliability" referred narrowly to P=8 > P=4 hallucinations; we now only mention this in Theorem 2 context.
>
> **re: LoRA vs. prefixes:** We suspect the difference is due to objective (CE loss vs. tail reliability) and technique (implicit vs. BT regularization). We agree prefixes sufficed for first-moment gains (ParScale) but show prefixes + LoRA + Barlow Twins are needed for second-moment gains. Expanded Table 4 shows prefixes alone achieve minimal decorrelation (D_spec=0.9991, 0.5% gain) while the full combination achieves D_spec=0.1400 and 4.9% gain. We'll be following up with LoRA rank sensitivity analysis.
>
> ---
>
> > **W3 + Q3:** "There is a missing corollary (line 144) in the theoretical analysis, and a missing figure (line 315) in the key results."
>
> Fixed. Missing corollary was a LaTeX error (proof now Lemma 1 + Theorem 1). Missing figure is now Figure 3 (correlation analysis, R²=0.237, p=0.002). We've also added Table 3, which provides a causal analysis (p < 0.001).
>
> ---
>
> > **W4:** "I don't like the claim in the abstract, 'ND-LoRA achieves 17.9% hallucination reduction with 1.12× training cost'. The 17.9% improvement occurs only in one subset of one dataset, and the rest of the results are in the range of 1-8% improvements. [...] I understand the choice made by the authors to showcase their strongest result, but still, the language is misleading"
>
> Revised to "up to 25.6% (and 14.6% on average)" stating both peak and average to both put best foot forward and be intellectually honest. (Post-submission work strengthened results as mentioned in global rebuttal.)
>
> ---
>
> We are deeply grateful for your expertise. The manuscript is much stronger thanks to your feedback. We would welcome any additional feedback — our goal is an honest, impactful paper.

---

> ### Author Response · Authors · 2025-12-01
>
> Dear Reviewer EyTf,
>
> We know you can't respond given the circumstances, but we just wanted to express our deep gratitude.
>
> Your push on novelty vs. ensemble theory was invaluable. Not only did it force us to articulate our precise contributions (the first formal tail bounds on P(hallucination) for ensembled LMs, distinct from classical E[loss] optimization), but your observation that our original formulation "just reduced to margins" pushed us to develop a much sharper theoretical link. Ultimately, we now concretely connect hallucinations in ensembled LMs to portfolio theory using Lipschitz smoothness and norm concentrations (overhauled Section 2), which beyond our paper we hope will begin connecting a critical research area (AI safety) and a rich research history (portfolio theory).
>
> We've uploaded [another revision](https://openreview.net/pdf?id=qQoLZ25tIs); in total, your concerns addressed:
> - **W1 + Q2 (Theoretical novelty):** Classical ensembles optimize E[loss]; we prove the first P(H) tail bounds in ensembled LMs, showing P(H) gains ~ 1/P with R²=0.962. Revised Section 2 concretely links rich portfolio theory field to LM hallucination via signal-noise modeling, Lipschitz decoding, and Lemma 1 exploiting shared-backbone structure to bound correlations.
> - **W2 (ParScale context):** Augmented Table 5 shows prefixes alone collapse (D=0.999, +0.5%); ND-LoRA achieves D=0.411, +12.8%. We also significantly expand on ParScale in Related Work, Ablations and Introduction. Different objectives require different techniques.
> - **W3 + Q3 (Missing content):** Fixed — Lemma 1 + Theorem 1, Figure 3 (p=0.002), Table 3 (p<0.001)
> - **W4 (Clarity):** Revised to "up to 25.6% (and 14.6% on average)"
> - **Q1 (Degradation):** Clarified scope; U-shaped behavior proven in Theorem 2
>
> Your rigor significantly strengthened both substance and contextualization. Regardless of the outcome, thank you.
>
> With appreciation,
> The Authors

---

### Official Review · Reviewer_3HiA · 2025-10-31

**Soundness:** 3
**Presentation:** 3
**Contribution:** 3
**Rating:** 6
**Confidence:** 1

**Summary:**

The paper proposes a theoretical framework linking neural diversity (quantified by cross-stream correlation) to hallucination probability, aiming to show that decorrelated neural representations directly lower the likelihood of false outputs. The paper proposes neural diversity as a novel scaling axis, complementary to model size and data volume, for reducing hallucination rates in small language models. Towards this end, the authors introduce ND-LoRA (Neural Diversity Low-Rank Adaptation), which integrates multiple independent LoRA adapters using the Barlow Twins regularization to encourage representational decorrelation among parallel computation streams. Experiments employing 8 benchmarks (TruthfulQA, HaluEval, MemoTrap, Natural Questions, TriviaQA, PopQA, Wikitext, Winogrande) with ND-LoRA are performed on Qwen2.5-0.5B with up to 8 streams showing that the proposed method is able to achieve an up to 17.9% hallucination reduction.

**Strengths:**

- Theoretical contributions reflect well in the experimental results (specifically, the relationship between reliability and neural diversity and the existence of an optimal neural diversity level in an U-shaped curve).
- Careful evaluation, accounting for parameter matching and statistical significance thresholds and confidence intervals, showing significant hallucination mitigation results at a minimal induced computation overhead.
- While the method specifically targets mitigating hallucinations, experimental results show it's not detrimental to overall performance.

**Weaknesses:**

- Experiments show high variability in optimal number of streams across tasks, implying repeated training or nontrivial hyperparameter search to find the sweet spot.
- Limited scale, with a small-scale backbone of 0.5B, 20M data tokens and context length of only 1k tokens (as far as I understand, many real-world hallucinations surface in longer chats where retrieval of the right snippet is harder, so external validity to long-context use remains uncertain).
- Concern about LoRA rank as a confound, given that several comparisons rely on parameter matching achieved through changing LoRA rank (making it 4 times smaller for instance in Table 1), as rank affects both expressivity and optimization dynamics and might confound conclusions about neural diversity.

As a small observation, there are five missing references at lines: 144, 315, 583, 600, 604.

**Questions:**

1. Theory assumes linearized properties and whitened features at the design layer. Might this be an over-assumption in practice? Have the authors considered any means of quantifying whether this occurs in real-world models?

2. Can an heuristic be found to choose P beforehand? Is there a way around training multiple models in order to try out multiple values for P?

---

> ### Author Response · Authors · 2025-11-17
>
> Dear Reviewer 3HiA,
>
> Thank you for your careful review and positive assessment. Your identification of the LoRA rank confound (W3) was particularly insightful — it's exactly the kind of technical rigor that strengthens experimental work and we're working on it. We respond to your questions inline.
>
> ---
>
> ## Weaknesses
>
> > **W1:** Experiments show high variability in optimal number of streams across tasks, implying repeated training or nontrivial hyperparameter search to find the sweet spot.
>
> This concern is reasonable (we probably would've had as reviewers too), but three clarifications:
>
> **(1) Zero hyperparameter tuning:** We did no hyperparameter tuning in the original submission. All tasks and P values used a single, shared hyperparameter set. The variability in optimal P emerges naturally from task characteristics, not search.
>
> **(2) Post-submission optimization strengthens patterns:** As summarized in the global rebuttal, post-submission hyperparameter optimization yielded stronger reductions (17.9% peak, 6.2% avg → 25.6% peak, 14.6% avg) while *clarifying* task-dependent structure: HaluEval peaks at P=4, TruthfulQA at P=2, knowledge tasks at P=1. Critically, we still use no task-specific tuning. Full benchmarks for all P∈{1,2,4,8} are in Appendix A.4.
>
> (answering jointly given overlap)
> > **Q2:** Can an heuristic be found to choose P beforehand? Is there a way around training multiple models in order to try out multiple values for P?
>
> **(3) Task-dependent optimality is a contribution, not a limitation:** We hypothesize different tasks require different precision-recall tradeoffs. Knowledge tasks (TriviaQA, NQ) favor P=1 to preserve recall; hallucination-focused tasks favor higher P: TruthfulQA (P=2) → HaluEval (P=4) → MemoTrap adversarial (P=8). Table 2 characterizes this landscape systematically. A simple heuristic: hallucination-sensitive tasks → higher P, knowledge-intensive → lower P.
>
> ---
>
> > **W2:** Limited scale (0.5B, 20M tokens, 1k context). Many real-world hallucinations surface in longer chats, so external validity to long-context remains uncertain.
>
> We transparently acknowledge these constraints in Section 7 (Limitations). However, we offer two counterpoints:
>
> **(1) SLMs choice is intentional:** They show disproportionate hallucination and increasing importance in agentic and edge use cases. Within this focus, 20M tokens is a *feature* — our technique is highly data-efficient.
>
> **(2) HaluEval explicitly tests longer contexts:** We agree that many real-world hallucinations surface in longer chats and, in fact, explicitly test [longer-context failure mode with HaluEval](https://arxiv.org/pdf/2305.11747). That we achieve our strongest gains on HaluEval (25.6%) indicates ND-LoRA addresses longer-context hallucinations particularly well.
>
> ---
>
> > **W3:** Concern about LoRA rank as a confound, given that several comparisons rely on parameter matching achieved through changing LoRA rank (making it 4 times smaller for instance in Table 1), as rank affects both expressivity and optimization dynamics and might confound conclusions about neural diversity.
>
> This is a very good point. As noted in global rebuttal, we'll be following up with a LoRA rank analysis. In the meantime, please also see (new) Figure 3 and (new) Table 2: our causal interventions (p < 0.001) and correlational studies (p = 0.002) show neural diversity directly drive hallucination reduction.
>
> > **W4:** Five missing references (lines 144, 315, 583, 600, 604).
>
> Fixed (see global rebuttal Section 3).
>
> ---
>
> ## Questions
>
> > **Q1:** Theory assumes linearized properties and whitened features at the design layer. Might this be an over-assumption in practice? Have the authors considered any means of quantifying whether this occurs in real-world models?
>
> **(1) Whitening enforced by design.** Although assumed for proof purposes in Lemma 1, in practice our Barlow Twins implementation explicitly whitens each stream's hidden states before computing cross-stream correlations.
>
> **(2) Local linearity is a weak assumption.** We do, however, substantively assume local linearity. This is a weak assumption and is used widely: saliency maps, adversarial examples, influence functions, and neural tangent kernels all rely on this.
>
> **(3)  Empirical validation supports both assumptions.** Figure 3 confirms the predicted diversity-hallucination relationship (R²=0.237, p=0.002), and our variance decomposition theory achieves R²=0.962 on empirical reliability variation (Figure 1). The idealized assumptions capture real behavior remarkably well.
>
> ---
>
> We hope these clarifications address your concerns. Thank you for your constructive feedback.

---

> ### Author Response · Authors · 2025-12-01
>
> Dear Reviewer 3HiA,
>
> We know you can't respond given the circumstances, but just wanted to say thank you regardless of the outcome. Your questions about LoRA rank confounds (W3) and heuristic for optimal P (Q2) pushed us to conduct a deeper analysis of both. We've updated both in our newest revision and, in fact, discovered that we could build a simple practical router that captures 97% of oracle performance.
>
> We've uploaded [another revision](https://openreview.net/pdf?id=qQoLZ25tIs); in total, your concerns addressed:
> - **W1 + Q2 (P variability, heuristic for P):** New Section 5.4 shows P=4 default captures 96% of oracle; simple router optimized for interpretability achieves 97%. Heuristic: Hallucination-sensitive tasks → higher P, knowledge-intensive → lower P. We release code.
> - **W2 (Context lengths):** HaluEval already tests longer-context distractions; our strongest gains (+25.6%) occur precisely here.
> - **W3 (LoRA confound):** New Appendix A.8 rules out capacity, alpha scaling and optimization dynamics as possible confounds.
> - **Q1 (Theory assumptions):** Augmented Figure 1 (R²=0.962) validates framework empirically.
>
> With appreciation,
> The Authors

---

### Author Response · Authors · 2025-11-17
**Substantially stronger core results, new mechanistic evidence & improved presentation**

Dear Reviewers,

We sincerely thank all three reviewers for their thoughtful and constructive feedback. We've uploaded a major new revision of our paper incorporating your feedback and other work done post-submission (e.g. 25.6% hallucination reduction, 4 new pieces of mechanistic evidence, and overhauled theory section):

**https://openreview.net/pdf?id=qQoLZ25tIs**

Your feedback -- especially R2's perspective on ensemble theory, R1's push about experimental rigor and R3's feedback about clarity -- helped us sharpen our theory, rigor and presentation. We're encouraged that reviewers recognize the potential of our approach — R1 noted "significant hallucination mitigation results" and R2 "the biggest strength of the paper is that the technique proposed works."

We look forward to your feedback.

---

## Major Changes in Revision 1
### 1. Substantially Stronger Core Results
- **Peak hallucination reduction: 17.9% → 25.6%** (HaluEval Summarization, Table 2); **Average: 6.2% → 14.6%** (6 benchmarks, Table 2)
- **Theory-empirics bridge: R² = 0.962** (revised Figure 1) — Theorems 1 & 2 explain 96.2% of empirical reliability variation seen across parallel configurations
- **Clearer, simpler, contrasted theory** (revamped Section 2): explicit ensemble/portfolio contrasts; signal-noise framing replacing confusing margins; all key results motivated and interpreted.

### 2. New Mechanistic & Statistical Analysis
- **Causality via interventions** (new Table 3, p<0.001 all tasks)
- **Correlation analysis** (new Figure 3, p=0.002): +0.1% neural correlation ↔ +3.8% hallucination
- **D_spec mediation** (augmented Table 4): ParScale collapses (0.9991), even with Barlow Twins (0.9988); LoRA + Barlow Twins essential (0.1400-0.3755)
- Module Ablations (new Appendix A.3): attention LoRA critical (-27.0% when removed)
- 4.6× more evaluations (183K samples >> 39K samples) with bootstrap CIs

### 3. Presentation & Other
- **Self-contained** (revised Abstract, Sections 1-2): all terms defined before use
- **Complete benchmarks** (new Appendix A.4): P ∈ {2,4,8} with parameter-matched baselines
- **Revamped Related Work** (revamped Section 6): comprehensive coverage of 1st/2nd-moment methods, ensemble/portfolio theory, redundancy reduction
- Full cost analysis (Appendix A.2): forward/backward/regularization breakdown
- Training efficiency: 2-4 hours → 30 minutes on A100
- Various fixes: missing references, LaTeX errors

## Core Contributions

Theoretically, we provide the first formal tail bounds on P(hallucination) for ensembled language models, reframing it as a second-moment reliability problem. Theorem 1 shows P(hallucination) ∝ 1/P when P components are perfectly decorrelated; we use portfolio theory (not ensemble theory) because of its focus on tail risk (vs. mean error). Beyond diminishing returns or overfitting from ensemble theory, we prove and validate (Figure 1, Theorem 2) that excessive parallelism can degrade diversity (and thus reliability) when correlation ρ(P) overwhelms 1/P scaling. Our predictions achieve R² = 0.962, explaining 96.2% of empirical variation — quantitative validation rare in hallucination research.

Empirically, we substantially improve on our previous result (see below) and show that ND-LoRA reduces hallucinations by up to 25.6% (14.6% average) at fixed budgets (Tables 1-2). To our knowledge, this is among the highest hallucination impacts in this ICLR cycle. We show that neural diversity mediates hallucinations via correlation (Figure 3), causality (Table 3, p<0.001), ablations (Table 4), and task-dependent optima (Appendix Tables 7-9).

---

In the next 1-2 weeks, we will follow up with additional improvements, e.g. analysis of potential LoRA confounds.

We are deeply grateful for your expertise. We value your continued feedback. If any aspect remains unclear, unconvincing, or could be improved, please let us know -- our goal is an honest, impactful contribution to LLM reliability.

With sincere appreciation,

The Authors

---

### Author Response · Authors · 2025-12-01
**25.6% Hallucination Reduction via Neural Diversity — First Formal Bounds, R²=0.962, All Concerns Addressed**

Dear Area Chair,

**tl;dr: ND-LoRA reduces hallucinations by up to 25.6%** (and 14.6% on average) — among the largest hallucination reductions this ICLR cycle, to our knowledge. We provide the **first formal tail bounds on P(hallucination) for ensembled LMs** and demonstrate that **neural diversity can explain 96.2% of observed empirical reliability gains** (R²=0.962). We achieve all of this via a simple, novel theoretical framework connecting portfolio theory and neural representations.

**Revised PDF: https://openreview.net/pdf?id=qQoLZ25tIs**

We thank all three reviewers for thoughtful feedback that significantly strengthened this paper. We appreciate you for jumping in during these extraordinary circumstances.

### Summary of Revisions
We've uploaded another major revision that contains several improvements. Summarizing all improvements to date:

| Key Concern           | Status       | Changes                                                                                         |
|--------------------------------|--------------|-------------------------------------------------------------------------------------------------|
| Clarity (R3)                   | Addressed    | Revised Abstract, Sections 1-2                                                                  |
| Theoretical Novelty (R2)       | Clarified    | First P(H) tail bounds for ensembled LMs, P(H) ~ 1/\|ensemble\| with R²=0.962, overhauled Section 2        |
| Causality (R3, R2)             | Demonstrated | New Table 3 (p<0.001), new Figure 3 (p=0.002), augmented Table 5                                |
| Practicality (R1)              | Implemented  | New Section 5.4, heuristic achieves 97% oracle, code will be released                               |
| LoRA Confounds (R1)            | Ruled out    | New Appendix A.8, Tables 10-11                                                                  |
| LaTeX Errors                   | Fixed        | Complete LaTeX cleanup                                                                          |

### Addressing Key Reviewer Concerns

* **Theoretical Novelty (R2):** Classical ensembles optimize E[loss]; we prove the first P(H) tail bounds in ensembled LMs, showing P(H) gains ~ 1/|ensemble| with R²=0.962. Revised Section 2 concretely links rich portfolio theory field to LM hallucination via signal-noise modeling, Lipschitz decoding, and Lemma 1 exploiting shared-backbone structure to bound correlations.

* **Causality (R3, R2):** We add three new lines of evidence: (i) Interventions perturbing diversity cause reliability drops (Table 3, p<0.001); (ii) +0.1% neural correlation ↔ +3.8% hallucination (Figure 3, p=0.002); (iii) Ablations mechanistically track diversity-hallucination relationship (Table 5).

* **Practicality (R1)**: New Section 5.4 shows P=4 default captures 96% of oracle; simple router optimized for interpretability achieves 97%. Heuristic: Hallucination-sensitive tasks → higher P, knowledge-intensive → lower P. We will release code.

* **Clarity & Rigor (R3):** The manuscript is now fully self-contained — all terms defined before use, all figures referenced, all notation systematically introduced, all LaTeX errors fixed.

* **LoRA Confounds (R1):** New Appendix A.8 rules out capacity, alpha scaling and optimization dynamics as possible confounds.

### Key Results
- **Hallucination reduction:** 25.6% peak (HaluEval Summarization) and 14.6% average (6 benchmarks)
- **Validated formal bounds:** P(H) ~ 1/P for decorrelated P, R²=0.962 against observed variation
- **Negligible cost:** +0.004% continued pretraining, 1.1× inference latency
- **Practicality:** P=4 captures 96% of oracle; simple router captures 97%

We believe this work makes a clear, timely contribution to LLM reliability and respectfully request the Area Chair's consideration.

---

### Meta-Review · Area_Chair_7DoC · 2025-12-16

**Summary:**

The paper studies how to reduce the rate of hallucinations of a small language model. By observing neural diversity, the authors propose  'ND-LoRA' to reduces hallucinations, without a significant increase in computation cost. However, the writting and presentation/structure of this paper is not good, which causes some potential issues on novelty. The current version is not ready for a decent submission. Besides, the authors also overclaim some stuffs on performance. Hence, I suggest to reject this paper.

**Reviewer Concerns:**

Reviewer EyTf complained about the unclear separation between this work and previous work, especially on the theoretical contribution. The comparison with ParScale is also missing. Besides, the paper has some overclaim on the performance with some missing figure. These issues were also pointed out by Reviewer pUug. The authors partially addressed these issues.

Reviewer 3HiA has quite a low confidence on the evaluation.

**Reviewer Scores:**

I'm not sure that Reviewer EyTf and pUug would change the score as the first impression is quite bad.

---

### Decision · Program_Chairs · 2026-01-26

Reject